# Youku Dense Caption: A Large-scale Chinese Video Dense Caption Dataset and Benchmarks

**Zixuan Xong**[1,3]   **Guangwei Xu**[3†] **Wenkai Zhang**   **Yuan Miao**[4]   **Xuan Wu**[4]   **Hai Lin**[1,2]
**Ruijie Guo**[3]   **Hai-Tao Zheng**[1,2‡]
[1]Shenzhen International Graduate School, Tsinghua University
[2]Pengcheng Laboratory    [3]Alibaba Cloud Computing     [4]Alibaba Group
`xzx22@mails.tsinghua.edu.cn`   `kunka.xgw@alibaba-inc.com`
`zhangwk0106@gmail.com`   `{miaoyuan.my, wx193834}@alibaba-inc.com`
`ngyygm@outlook.com`   `ruijie.guo@alibaba-inc.com`
`zheng.haitao@sz.tsinghua.edu.cn`

## Abstract

With the explosive growth of video content, video captions have emerged as a crucial tool for video comprehension, significantly enhancing the ability to understand and retrieve information from videos. However, most publicly available dense video captioning datasets are in English, resulting in a scarcity of large-scale and high-quality Chinese dense video captioning datasets. To address this gap within the Chinese community and to promote the advancement of Chinese multi-modal models, we develop the first, large-scale, and high-quality Chinese dense video captioning dataset, named Youku Dense Caption. This dataset is sourced from Youku, a prominent Chinese video-sharing website. Youku Dense Caption includes 31,466 complete short videos annotated by 311,921 Chinese captions. To the best of our knowledge, it is currently the largest publicly available dataset for fine-grained Chinese video descriptions. Additionally, we establish several benchmarks for Chinese video-language tasks based on the Youku Dense Caption, including retrieval, grounding, and generation tasks. Extensive experiments and evaluations are conducted on existing state-of-the-art multi-modal models, demonstrating the dataset's utility and the potential for further research. The dataset is publicly available at https://github.com/OpenSearch-AI/Youku-Dense-Caption.

## 1 Introduction

The proliferation of the internet and mobile devices has precipitated an exponential increase in the volume of video content (Wang & Gao, 2020). Across social media platforms, video-sharing websites, and online education and entertainment applications, video has emerged as a predominant medium for information dissemination and consumption (Levy, 2021). However, this surge in video content presents a significant challenge in efficiently understanding and retrieving information from videos (Dong et al., 2022b). Among the various auxiliary tools available, video captions have garnered considerable attention for their pivotal role in enhancing video content comprehension and facilitating information retrieval (Gernsbacher, 2015; Perego et al., 2010).

Currently, the majority of publicly available dense video captioning datasets are predominantly in English, resulting in a notable scarcity of resources for non-English languages, particularly Chinese. This linguistic gap not only constrains the user experience for Chinese-speaking audiences but also impedes the development and optimization of multi-modal models tailored to Chinese video content (Li et al., 2019; Singh et al., 2020).

To address this critical gap and foster the advancement of Chinese multi-modal models, we introduce the Youku Dense Caption dataset, the first large-scale, high-quality Chinese dense video cap-

---

[†]Project Lead.
[‡]Corresponding author: zheng.haitao@sz.tsinghua.edu.cn

Table 1: Statistics of Youku Dense Caption and its comparison with existing video-language datasets. *Avg. Len* denotes the average segment length, referring to the mean length of the segments designated by each annotation.

| Datasets | Type | Language | # Videos | # Text | Avg. Len | Domain |
|---|---|---|---|---|---|---|
| MSR-VTT (Xu et al., 2016) | video captioning | EN | 7k | 200k | 14.8s | open |
| MSVD (Chen & Dolan, 2011) | video captioning | EN | 2k | 85k | 7s | open |
| VATEX (Wang et al., 2019b) | video captioning | EN/CH | 41k | 826k | 10s | open |
| Youku-mPLUG (Xu et al., 2023) | video captioning | CN | 80k | 80k | 54.2s | open |
| ActivityNet Captions (Krishna et al., 2017) | dense video captioning | EN | 20k | 100k | 36s | open |
| YouCook2 (Zhou et al., 2018) | dense video captioning | EN | 2k | 15k | 7.7s | cook |
| ViTT (Huang et al., 2020) | dense video captioning | EN | 8k | 12k | - | open |
| Youku Dense Caption | dense video captioning | CN | 31k | 311k | 8.1s | open |

tioning dataset, meticulously designed to meet the needs of Chinese video content comprehension and information retrieval. This dataset is sourced from Youku, one of China's leading video-sharing platforms, and comprises 31,466 complete short videos annotated with 311,921 Chinese captions. This makes it the largest and most detailed publicly available dataset for fine-grained descriptions of Chinese video content, thereby providing a substantial resource for Chinese video-language processing research.

In addition to offering a comprehensive dataset, we establish several benchmarks for Chinese video-language tasks based on the Youku Dense Caption dataset. These tasks include video retrieval, grounding, and caption generation. These benchmarks not only provide rigorous setup pipeline for the objective evaluation of existing multi-modal models, but also guide future research and development directions in this domain.

To validate the utility of the Youku Dense Caption dataset, we conduct extensive experiments and evaluations using state-of-the-art multi-modal models. The results of these experiments demonstrate the dataset's significant impact on improving model performance, including video retrieval and caption generation. Through this research, we underscore the potential of the Youku Dense Caption dataset in advancing the field of Chinese video-language development.

Our main contributions are as follows:

• We introduce the Youku Dense Caption dataset, the largest and fully human-annotated Chinese dense video captioning dataset, with 31,466 short videos and 311,921 Chinese captions.

• We establish several benchmarks for Chinese video-language tasks, including video retrieval, grounding, and generation, providing standard evaluation metrics for multi-modal models.

• We validate the effectiveness of the dataset through extensive experiments, demonstrating its significant impact on enhancing the performance of multi-modal model generation and retrieval.

## 2 RELATED WORK

**Dense Video Captioning Dataset**   Dense video captioning (DVC) datasets represent a specialized subset of video-captioning resources, distinguishing themselves from traditional datasets by focusing on the nuanced task of narrating multiple events within a video rather than generating a single overarching description. The DVC datasets discussed, such as MSR-VTT (Xu et al., 2016), MSVD (Chen & Dolan, 2011), YouCook2 (Zhou et al., 2018), VATEX (Wang et al., 2019b), ActivityNet Captions (Krishna et al., 2017), YouMakeup (Wang et al., 2019a), and ViTT (Huang et al., 2020), provide rich annotations that include event-level details and diverse categories, enabling models to capture and describe events sentence by sentence. These datasets vary in size, scope, and linguistic diversity. For instance, MSR-VTT comprises around 10,000 videos across various domains, while VATEX offers multilingual annotations for over 41,000 clips. Moreover, datasets like YouCook2 (Zhou et al., 2018) highlight domain specificity, focusing on cooking videos, thus facilitating tasks like recipe generation and video summarization. Collectively, these resources advance research in video understanding, natural language processing, and the development of sophisticated DVC models capable of effective event localization and description.

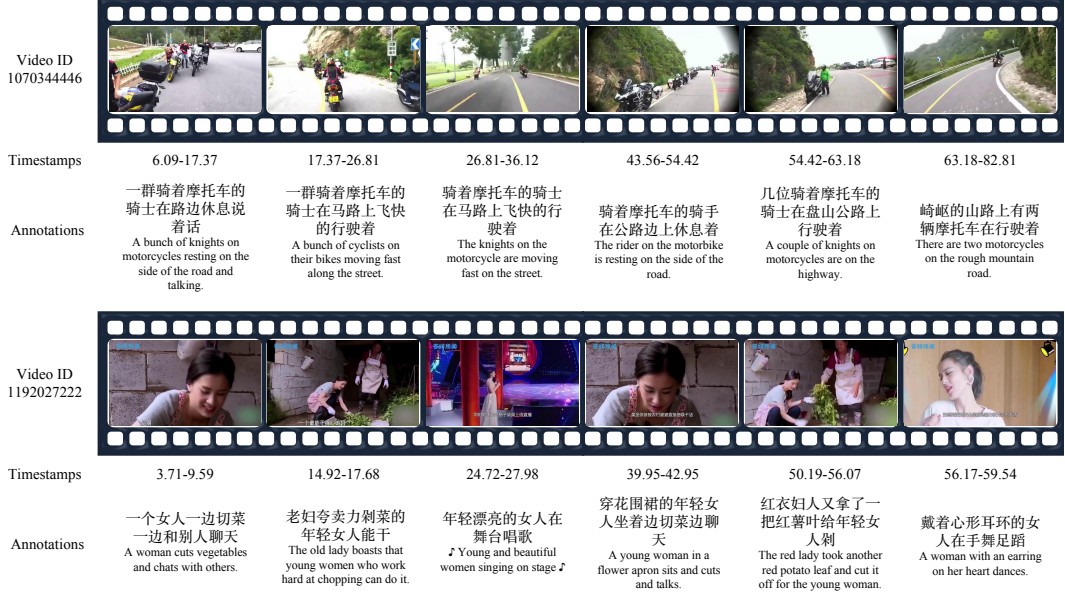

Figure 1: Sample videos and annotations from the Youku Dense Caption dataset. We randomly select two videos and sample six frames from each, displaying their respective timestamps and corresponding Chinese text annotations. Additionally, we provide English translations of the annotations using the open-source tool OPUS-MT (Tiedemann et al., 2023; Tiedemann & Thottingal, 2020).

**Video-Language Downstream Benchmarks**   The related works (Wang et al., 2024b; Xu et al., 2023) on video-language downstream benchmarks primarily focus on evaluating and enhancing the performance of tasks related to video content description and retrieval, encompassing various international challenges and competitions. For instance, the Large Scale Movie Description Challenge (LSMDC) (Rohrbach et al., 2017) is based on a movie video dataset and aims to generate both single and multiple sentences that describe video segments, utilizing human and automated evaluation methods to assess the quality of the generated text. Similarly, the Video-to-Language challenge, based on the MSR-VTT dataset, requires participants to generate complete sentences that describe video content, combining both automated and manual evaluations for scoring. Additionally, the TREC Video Retrieval Evaluation (TRECVID) (Awad et al., 2023) explores the matching and generation tasks between video and text descriptions. Finally, the VATEX Video Description Challenge focuses on multilingual video descriptions, encouraging the use of video information as context to enhance translation quality. These benchmark studies promote advancements in video content understanding and natural language description by proposing diverse tasks and evaluation standards.

## 3   YOUKU DENSE CAPTION DATASET

To address the gap of fine-grained annotated datasets in the Chinese community, we present the first large-scale dense video captioning dataset with detailed Chinese annotations. The dataset comprises 31,466 videos, segmented into 311,921 clips, with a cumulative duration of 748.96 hours. Each video has an average duration of 85.68 seconds, while the average clip length is 8.1 seconds. On average, each video contains 9.9 annotations, with each annotation consisting of 17.9 characters. The entire process, from data cleaning to the generation of Chinese annotations, was meticulously carried out manually to ensure the highest quality of data.

### 3.1   DATA SOURCES

The dataset is constructed to meet the following requirements: 1) it should cover the most common video themes; 2) the duration of the videos should be no less than one minute to ensure that the content is meaningful. Based on these requirements, the raw videos in the dense caption dataset are evenly sampled from the Youku-mPLUG dataset based on 11 major categories and 84 subcategories,

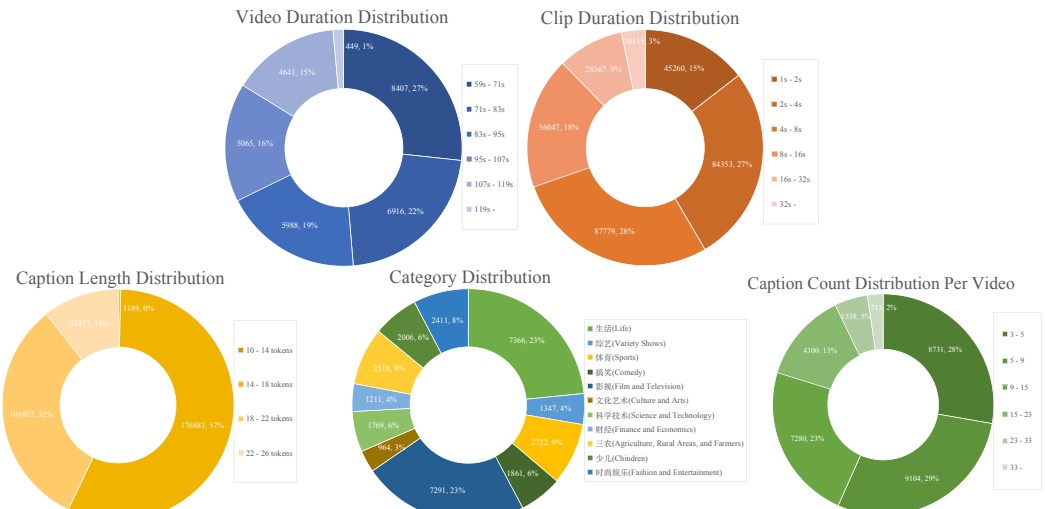

Figure 2: Statistics of Youku Dense Caption. Please note that in *A, B%*, A represents the exact quantity, while B represents the corresponding percentage.

with this categorization derived from the video tags. The Youku-mPLUG dataset contains 10 million videos collected according to strict safety, diversity, and quality criteria. In addition, our annotators are instructed to exclude any videos that presented difficulties in annotating, such as those that are unclear, split screens, illustrations, image carousels, or meaningless effects, ensuring that the videos in the dataset are of high quality and rich in event information. More annotation details are in the Appendix.

## 3.2 STATISTICS AND FEATURES

We now present the statistics of the Youku Dense Caption dataset and compare them with other similar datasets. Fig. 2 summarizes the most notable aspects of the Youku Dense Caption dataset concerning both its visual and linguistic content.

### 3.2.1 BASIC STATISTICS

The Youku Dense Caption dataset is characterized by its diversity and richness, making it a valuable resource for research in video understanding, caption generation, and natural language processing.

**Extensive Category Coverage.** The dataset includes 11 coarse-grained categories (e.g., Life, Variety Shows, Finance and Economics, Agriculture, Children, Fashion and Entertainment) and 84 fine-grained subcategories (e.g., Film and Television includes movies, TV shows, anime, etc.). This wide range ensures comprehensive coverage of video types across open domains. **Balanced Video Duration.** 99% of the videos have durations between 1 to 2 minutes, ensuring rich content and feasibility for fine-grained annotations. The duration segments are relatively balanced, each comprising about 20% of the dataset, providing a good representation of different video lengths. **Short Clip Focus.** Predominantly featuring clips ranging from 2 to 32 seconds, the dataset is well-suited for short video content analysis and processing. The uniform distribution of clip durations aids in model generalization across different clip lengths. **Consistent Caption Lengths.** Captions are primarily between 10 to 26 tokens, with the majority in the 18 to 22 token range. This consistency supports dense captioning and short text generation tasks, helping maintain stable model performance during training. **Rich Caption Information.** Each video contains 3 to 33 captions, providing ample data for dense captioning and video content understanding research. This layered distribution facilitates studies on video comprehension and generation under varying caption densities.

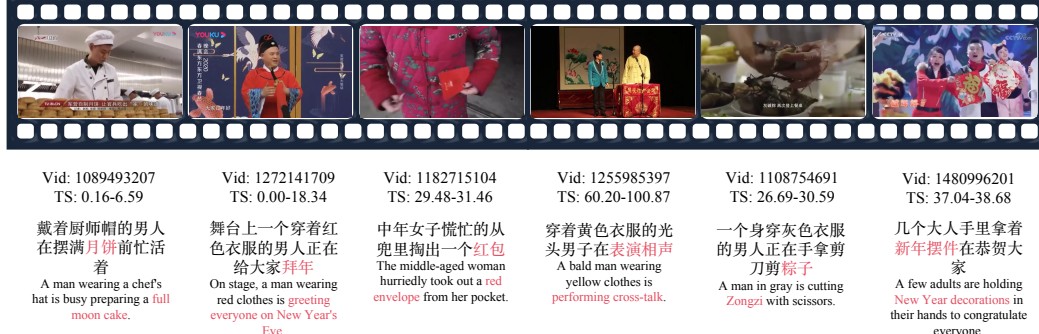

Vid: 1089493207
TS: 0.16-6.59

Vid: 1272141709
TS: 0.00-18.34

Vid: 1182715104
TS: 29.48-31.46

Vid: 1255985397
TS: 60.20-100.87

Vid: 1108754691
TS: 26.69-30.59

Vid: 1480996201
TS: 37.04-38.68

戴着厨师帽的男人
在摆满月饼前忙活
着
A man wearing a chef's
hat is busy preparing a full
moon cake.

舞台上一个穿着红
色衣服的男人正在
给大家拜年
On stage, a man wearing
red clothes is greeting
everyone on New Year's
Eve.

中年女子慌忙的从
兜里掏出一个红包
The middle-aged woman
hurriedly took out a red
envelope from her pocket.

穿着黄色衣服的光
头男子在表演相声
A bald man wearing
yellow clothes is
performing cross-talk.

一个身穿灰色衣服
的男人正在手拿剪
刀剪粽子
A man in gray is cutting
Zongzi with scissors.

几个大人手里拿着
新年摆件在恭贺大
家
A few adults are holding
New Year decorations in
their hands to congratulate
everyone.

Figure 3: Examples of culture unique to China in the dataset. *Vid* indicates video ID, and *TS* stands for timestamp. The red text highlights instances of unique and iconic terms in Chinese culture.

### 3.2.2 CHINESE CHARACTERISTICS

Firstly, there exist significant differences in cultural background and contextual understanding between Chinese and English-speaking communities. Unique elements of Chinese culture, such as the tea ceremony, Mid-Autumn Festival moon gazing, and Chinese New Year greetings, are rarely encountered in public English datasets but are quite common in Chinese contexts. This cultural disparity is evident not only in the distribution of annotated texts but also in the content of videos. For detailed cultural explanations, please refer to Appendix G.

As emphasized by the widely used Chinese CLIP (Yang et al., 2022), bridging the gap in annotations and video content caused by cultural differences is the core value of our dataset. Understanding and bridging these cultural differences are crucial for accurate translation and content representation. By highlighting these unique cultural elements, our dataset not only better reflects the actual needs of Chinese users but also provides greater accuracy and cultural relevance in the distribution of annotations and video content.

## 4 BENCHMARK SETUP

### 4.1 PARTIALLY RELEVANT VIDEO RETRIEVAL

An untrimmed video is considered partially relevant to a given textual query if it includes a segment that pertains to the query. Partially Relevant Video Retrieval (PRVR) (Dong et al., 2022a) aims to identify and retrieve these partially relevant videos from a vast collection of untrimmed videos. Existing English benchmarks, such as TV Show Retrieval (Lei et al., 2020), ActivityNet Captions, and Charades-STA (Gao et al., 2017), directly use segment-level annotated texts as queries to search within video. This approach overlooks the possibility of similar segments and annotations existing across different videos, which leads to these similar annotations matching with multiple videos rather than being specific to a single video. Furthermore, these similar annotations cause nearly repetitive retrieval processes during evaluation, exacerbating the evaluation bias of the benchmark. To address this issue, we post-process the data to establish a Chinese PRVR benchmark. This benchmark ensures that a single query corresponds to multiple videos without repetitive query retrieval, thereby enabling a more debiased evaluation.

First, we extract the segment-level annotation and video ID pairs to form the pairs $[t_i, v_i]$. Then, we aggregate similar texts using the xiaobu-embedding model (Huang et al., 2024; Sun et al., 2020), which has achieved state-of-the-art results in several Chinese retrieval tasks (Muennighoff et al., 2023). For each annotation text $t_i$, we calculate its embedding and then create a set $T_i$ of all annotation texts with an embedding similarity greater than 90%, along with a corresponding set $V_i$ of video IDs. Thus, for each annotation text $t_i$, there is a corresponding set $V_i$, indicating that all videos in $V_i$ contain segments related to $t_i$.

In the second step, we focus on removing duplicate annotation texts to avoid redundancy. We employ a method similar to Non-Maximum Suppression (NMS), commonly used in object detection tasks.

---

**Algorithm 1** Cross-Video PRVR Benchmark Setup

---

1: **Input:** Dataset $D = \{(t_i, v_i), (t_j, v_i), \cdots, (t_k, v_j)\}$, $t_i$ is a segment-level annotation.
2: **Output:** Benchmark $B = \{(t_i, V_i), \cdots, (t_j, V_j)\}$, $V_i$ is a set of video ids.
3: **for** each annotation text $t_i$ **do**
4:     embed$(t_i)$ = Text Encoder$(t_i)$.
5:     Create set $T_i$ of all annotation texts with embedding similarity $> 0.9$ to $t_i$:

$$T_i = \{t_j \mid \text{sim}(\text{embed}(t_i), \text{embed}(t_j)) > 0.9\}$$

6:     Create the corresponding set $V_i$ of video IDs from pairs $(t_j, v_j)$ where $t_j \in T_i$:

$$V_i = \{v_j \mid (t_j, v_j) \in \{(t_i, v_i)\} \text{ and } t_j \in T_i\}$$

7: **end for**
8: **for** each pair of sets $(V_i, V_j)$ **do**
9:     **if** len$(V_i) > 1$ and len$(V_j) > 1$ **then**
10:         Calculate Intersection over Union (IoU):

$$\text{IoU}(V_i, V_j) = \frac{|V_i \cap V_j|}{|V_i \cup V_j|}$$

11:         **if** IoU$(V_i, V_j) > 0.7$ **then**
12:             Remove $(t_j, V_j)$ if len$(V_i) >$ len$(V_j)$, else remove $(t_i, V_i)$.
13:         **end if**
14:     **end if**
15: **end for**
16: **Output:** Remaining pairs $(t_i, V_i)$.

---

First, we calculate the Intersection over Union (IoU) between each pair of sets $V_i$ and $V_j$. The IoU is defined as the size of the intersection divided by the size of the union of the two sets:

If the IoU between $V_i$ and $V_j$ exceeds 0.7, we consider the retrieval process for $t_i$ and $t_j$ to be redundant. To resolve this redundancy, we compare the sizes of the sets $V_i$ and $V_j$. If len$(V_i) >$ len$(V_j)$, we retain the larger set $V_i$ and remove the annotation $t_j$ along with its corresponding set $V_j$. This iterative process ensures that we retain the most representative and comprehensive annotation sets, thereby reducing redundancy.

As a result of these steps, we build a Chinese PRVR benchmark based on Youku Dense Caption, which consists of 28,988 queries and 3,185 target videos, with an average of 4.96 related videos per query. We set two evaluation metrics: one is the Top-k Accuracy, where retrieving any one of the related videos within the top k results is considered meeting the requirements, and the other is the Mean Average Precision, which measures the proportion of related videos in the top len$(V_i)$ results.

## 4.2 GROUNDING AND GENERATION

In the tasks of temporal grounding and caption generation, existing annotations of text and video segments face two main issues. Firstly, the annotated text for certain videos may contain a large amount of repetitive information due to the lack of event changes. Such annotations are usually ambiguous and do not provide informative descriptions of the video segments. Secondly, the scene changes within video segments may not be significant. Even though the video is divided into different timestamps, the internal scene changes within the segments are minimal, resulting in repetitive and monotonous annotations. This monotony reduces both the diversity and objectivity of the evaluation, making these video segments unsuitable for use as evaluation data in tasks like temporal grounding and generation.

To address these issues, we propose two strategies to improve the quality of benchmark datasets: First, we calculate the internal consistency of annotated texts. Specifically, we calculate the self-BLEU (Zhu et al., 2018) value for all annotated texts within each video. If the self-BLEU value of a video exceeds 0.15, the internal consistency of the annotated texts is considered too high, and the video is subsequently filtered out.

The self-BLEU value for an annotated text $s_i$ is calculated as follows:

$$\text{Self-BLEU}(s_i) = \text{BLEU}(s_i, \{s_1, s_2, \ldots, s_{i-1}, s_{i+1}, \ldots, s_n\}) \tag{1}$$

where $s_i$ is the annotated text being evaluated, and $\{s_1, s_2, \ldots, s_{i-1}, s_{i+1}, \ldots, s_n\}$ represents the set of all other annotated texts within the same video, excluding $s_i$.

The overall self-BLEU for a video is then calculated as the average of the self-BLEU values for all annotated texts $s_i$ in that video:

$$\text{Self-BLEU} = \frac{1}{n} \sum_{i=1}^{n} \text{Self-BLEU}(s_i) \tag{2}$$

where $n$ is the total number of annotated texts within the video.

Second, we use the color histogram correlation of video segments to assess the degree of scene change between video segments. We perform color histogram analysis on each video segment and calculate the histogram correlation between adjacent segments. Specifically, the formula for calculating histogram correlation (Correlation) is as follows:

$$\text{Correlation}(H_1, H_2) = \frac{\sum_{i=1}^{n}(H_1(i) - \bar{H}_1)(H_2(i) - \bar{H}_2)}{\sqrt{\sum_{i=1}^{n}(H_1(i) - \bar{H}_1)^2}\sqrt{\sum_{i=1}^{n}(H_2(i) - \bar{H}_2)^2}} \tag{3}$$

where $H_1$ and $H_2$ represent the color histograms of two video segments, $H_1(i)$ and $H_2(i)$ represent the value of the $i$th bin in the histograms, and $\bar{H}_1$ and $\bar{H}_2$ represent the mean values of $H_1$ and $H_2$, respectively. The mean values are calculated as follows:

$$\bar{H}_1 = \frac{1}{n} \sum_{i=1}^{n} H_1(i), \quad \bar{H}_2 = \frac{1}{n} \sum_{i=1}^{n} H_2(i) \tag{4}$$

If the color histogram correlation of a video segment exceeds 0.5, the scene change in the video is considered insignificant, and the video is subsequently filtered out. In the end, we obtain 1,872 videos and a total of 20,099 annotations, averaging 10.73 annotations per video.

The evaluation metrics are based on existing work (Soldan et al., 2022; Pan et al., 2023). For grounding tasks, we use the R@N and IoU=M calculation methods, which assess the top-N accuracy under the condition that the IoU is greater than M for the retrieved timestamps. For generation tasks, mainstream evaluation metrics such as BLEU (Papineni et al., 2002), METEOR (Lavie & Agarwal, 2007), CIDEr (Vedantam et al., 2015), and ROUGE-L (Lin, 2004) are used for a comprehensive assessment of the generated text. In addition, we also include the Bert score (Zhang et al., 2020) to measure the semantic similarity of the generated text to alleviate the evaluation bias caused by different generation styles.

## 5 EXPERIMENTS

In this section, we comprehensively evaluate several state-of-the-art video multi-modal large models on three benchmarks within the Youku Dense Caption dataset. We present more detailed introductions to the baselines and implementation details in the Appendix.

**Retrieval Performance** In this experiment, we select three categories of models: video retrieval models, video generation models, and CLIP models. The rationale behind choosing these three types is to comprehensively evaluate the performance of different multi-modal large models on Chinese video datasets and understand the impact of language differences. Given that the Youku dense caption dataset is in Chinese, and most existing large-scale video models, such as InternVideo (Wang et al., 2022; 2024c) and ViCLIP (Wang et al., 2024b), are primarily trained on English corpora, we translate the annotations into English using OPUS-MT to conduct bilingual evaluations in both Chinese and English.

Table 2: Evaluation results of the retrieval benchmark. We evaluate the existing various video models, image-text models, and generative models using only zero-shot methods. For specific experimental details, please refer to the Appendix.

| Models | Language | text2video | | | video2text | | |
|---|---|---|---|---|---|---|---|
| | | Top 1 Acc | Top 5 Acc | Top 10 Acc | Top 1 Acc | Top 5 Acc | Top 10 Acc |
| ViCLIP (Wang et al., 2024b) | English | 26.78 | 47.53 | 56.33 | 56.38 | 76.95 | 84.39 |
| InternVideo2 (Wang et al., 2024c) | English | 26.56 | 47.46 | 57.17 | 47.87 | 70.21 | 79.07 |
| ViCLIP | Chinese | 0.91 | 4.71 | 7.92 | 2.12 | 8.86 | 12.41 |
| InternVideo2 | Chinese | 1.31 | 3.90 | 6.94 | 1.77 | 4.60 | 7.80 |
| mPLUG-Owl3 (Ye et al., 2024) | Chinese | 2.04 | 6.57 | 10.55 | 0.35 | 2.48 | 3.19 |
| CLIP Radford et al. (2021) | English | 1.05 | 4.34 | 8.18 | 2.12 | 9.92 | 13.82 |
| DFN5B-CLIP (Fang et al., 2024) | English | 6.61 | 15.49 | 24.00 | 16.31 | 36.87 | 47.51 |
| Chinese-CLIP (Yang et al., 2022) | Chinese | **31.78** | **55.49** | **65.72** | **41.13** | **64.53** | **70.92** |

For the video retrieval models, when comparing the retrieval performance of ViCLIP and InternVideo2 on Chinese and English datasets, it is evident that these models perform poorly on Chinese video retrieval tasks. This underperformance is primarily due to the lack of training on Chinese corpora.

Regarding the video generation model, mPLUG-Owl3, which only supports multi-modal generation tasks, we adapted it for the retrieval task by performing pooling on the token embeddings encoded by its model to serve as its multi-modal representation. Despite this adaptation, mPLUG-Owl3 does not exhibit effective retrieval performance in its encoded embeddings. This is because mPLUG-Owl3 lacks specific retrieval training, which is crucial for such tasks.

Lastly, for the CLIP models, since there are no native Chinese video large models available for comparison against English video large models like ViCLIP, we compare the retrieval performance of Chinese and English CLIP models of the same size. The Chinese-encoded model shows a significant advantage over the English model when evaluated on Chinese and translated English datasets. This finding indirectly highlights the language differences between Chinese and English, emphasizing the importance of language-specific training for optimal performance.

Overall, these experiments underscore the necessity of training multimodal models on diverse linguistic datasets to ensure robust performance across different languages.

**Grounding Performance** In the grounding task, due to the lack of temporally related tasks during the pre-training phase of the pre-trained multi-modal large model (Feng et al., 2023), multi-modal large models cannot be directly applied to downstream tasks such as temporal localization. Therefore, following Wang et al. (2024c), we use these pre-trained models as feature extractors to encode multi-modal features and input them to Moment-DETR (Lei et al., 2021) for training, resulting in the Tab. 3.

Table 3: Evaluation results of the grounding benchmark. In the evaluation, we use the same-dimensional CLIP and Chinese CLIP for comparison to ensure the capacity of the temporal localization model remains consistent.

| Features | Dim | Language | R1@0.5 | R1@0.7 | mAP@0.5 | mAP@0.75 | mAP@avg |
|---|---|---|---|---|---|---|---|
| CLIP | 512 | English | 6.38 | 2.16 | 11.70 | 3.24 | 4.35 |
| ViCLIP | 768 | English | 6.23 | 2.17 | 11.40 | 3.31 | 4.30 |
| Chinese CLIP | 512 | Chinese | **13.89** | **4.95** | **20.10** | **5.91** | **7.76** |

Due to the lack of Chinese-supported video-language encoding models, we only employ ViCLIP and CLIP-type models that perform well in retrieval tasks. It can be observed that, under the same encoding dimensions, Chinese-CLIP still demonstrates a greater advantage compared to CLIP.

**Generation Performance** For generative tasks, most existing multi-modal large models rely on pre-trained large language models, which generally support multilingual generation. Therefore, we do not perform additional language processing for this task. For different models, we use the same Chinese prompt *Describe this video in one sentence.*.

Table 4: Evaluation results of the generation benchmark. We directly segment the video using timestamps and generate descriptions for the video segments using these models.

| Models | Vision Encoder | Language Decoder | BLEU-4 | METEOR | Rouge-L | CIDEr | Bert Score |
|---|---|---|---|---|---|---|---|
| InternVideo2-Chat-8B | Internvideo2-1B | Mistral-7B | 0.35 | 6.92 | 7.25 | 4.97 | 58.69 |
| MiniCPM-V-2_6 | SigLip-400M | Qwen2-7B | 0.57 | 12.86 | 12.14 | 3.66 | 62.72 |
| InternVL2-8B | Intern-ViT-6B | InternLM2.5-7B | 1.09 | 13.62 | 13.20 | 9.19 | 65.34 |
| Qwen2-VL-7B-Instruct | DFN | Qwen2-7B | **1.51** | **13.84** | **15.10** | **19.32** | **66.68** |

As shown in Tab. 4, InternVL2-8B (Chen et al., 2023; Jiang et al., 2023), MiniCPM-V-2.6 (Yao et al., 2024; Zhai et al., 2023; Yang et al., 2024) and Qwen2-VL-7B-Instruct (Bai et al., 2023; Wang et al., 2024a) exhibit significant advantages in video captioning. These models have undergone extensive training for image-text generation, which enhances their performance in this domain. In video processing, large models typically perform frame extraction, converting videos into multiple images for captioning tasks. This approach allows video captioning to effectively leverage the strong image-text captioning capabilities of these models.

# 6    ABLATION STUDY

In this section, we experimentally evaluate the quality of our dataset as training data and how it can enhance the model's performance on existing benchmarks within the community. Specifically, we fine-tune multi-modal models such as ViCLIP and Qwen2-VL on datasets with Chinese video annotations, including VATEX and Youku-mPLUG. Additionally, we introduce our proposed Youku Dense Caption (YDC) dataset as extra training data during the fine-tuning process and evaluate the performance on the corresponding benchmarks to demonstrate the effectiveness of our dataset for training.

**Improving Generation Performance with YDC**    We select the VATEX dataset with Chinese annotations for auxiliary validation because it offers ten semantically similar annotations for each video. Furthermore, due to stylistic differences across various domains in generation tasks, we evaluate both the VATEX validation set and the generation benchmark on YDC, which are of nearly the same scale. This dual evaluation aims to assess the cross-domain generalization capabilities of the training results.

Table 5: Experiments on achieving data augmentation in generation tasks using YDC data.

| Training Set | | VATEX Val | | | | Generation Benchmark on YDC | | | |
|---|---|---|---|---|---|---|---|---|---|
| % VATEX | % YDC | BLEU-4 | METEOR | ROUGE-L | CIDEr | BLEU-4 | METEOR | ROUGE-L | CIDEr |
| 0% | 0% | 16.00 | 22.52 | 39.01 | 33.06 | 0.94 | 9.39 | 12.57 | 15.87 |
| 100% | 0% | 28.16 | 30.17 | 48.05 | 53.12 | 2.41 | 14.06 | 18.36 | 26.88 |
| 50% | 50% | 27.71 | 29.81 | 47.72 | 51.52 | 4.71 | 17.31 | 25.23 | 42.41 |
| 0% | 100% | 22.77 | 26.50 | 43.74 | 35.64 | 4.86 | 17.50 | 25.51 | 43.06 |
| 100% | 50% | 28.42 | 30.14 | 47.98 | 52.82 | 4.63 | 17.34 | 25.28 | 41.51 |
| 100% | 100% | **29.57** | **30.49** | **48.53** | **54.06** | **4.91** | **17.54** | **25.68** | **44.32** |

We conduct LoRA fine-tuning (Hu et al., 2022) based on the Qwen2-VL-2B model and compared different experimental results by adjusting the proportion of training data sources. Specifically, on one hand, we control the total amount of training data to remain roughly consistent and adjust the proportion of the VATEX and Youku datasets for training. On the other hand, we gradually add the YDC dataset while utilizing the full VATEX dataset to observe the data augmentation effect. Note that the VATEX training data contains 250k annotated texts, while the training set filtered and divided from the Youku Dense Caption for the generation task contains 278k video-text pairs, with similar data volumes.

During the experiments, we find that models trained only on VATEX or only on Youku Dense Caption did not achieve ideal results on the evaluation set of the other domain. This indicates that there are significant differences between data from different domains, and training data from a single source cannot achieve generalization across different domains.

To overcome this limitation, we attempt to mix the training data of VATEX and Youku Dense Caption. The results show that as the proportion of YDC training data increased, the Qwen2-VL model eventually achieved state-of-the-art results on both evaluation sets. This demonstrates that our proposed YDC dataset, similar to the VATEX dataset, can bring significant data augmentation effects.

Additionally, we provided additional test results on the youku-mplug validation set under the same training set scale. The results demonstrate that, between datasets of comparable size, training on the Youku Dense Caption dataset achieves superior performance compared to Vatex. Additionally, to mitigate potential effects of data distribution bias, we evaluated the model on the Youku-mplug dataset, which represents a different distribution from the training data.

Table 6: Experiments on out-of-distribution (OOD) performance.

| Training Data | Youku-mPLUG | BLEU-4 | METEOR | ROUGE-L | CIDEr |
|---|---|---|---|---|---|
| Vatex | Validation | 3.08 | 12.77 | 19.53 | 29.53 |
| Youku Dense Caption | Validation | 3.46 | 13.54 | 22.76 | 31.55 |

**Improving Retrieval Performance with YDC**   Here, we explore the contribution of the YDC data to the video-text retrieval task, using the Youku-mPLUG dataset as the base data. To ensure no overlap between Youku-mPLUG and YDC, we deduplicate the Youku-mPLUG data using file hashing and several video features. We select ViCLIP for fine-tuning, which achieves state-of-the-art results in English. The Youku-mPLUG retrieval training data contains 36k video-text pairs, while the YDC dataset's retrieval training data amounts to 278k pairs. Therefore, 13% of the YDC training data in our experiments is equivalent in size to the Youku-mPLUG training data.

Table 7: Experiments on achieving data augmentation in retrieval tasks using YDC data.

| Training Set | | Testing Set | video2text | | | text2video | | |
|---|---|---|---|---|---|---|---|---|
| % Youku-mPLUG | % YDC | Youku-mPLUG | R1 | R5 | R10 | R1 | R5 | R10 |
| 0% | 0% | Validation | 1.03 | 2.94 | 4.78 | 0.40 | 1.32 | 2.24 |
| 100% | 0% | Validation | 2.42 | 9.86 | 16.55 | 1.90 | 8.36 | 14.12 |
| 0% | 13% | Validation | 0.46 | 2.30 | 4.09 | 0.34 | 1.55 | 3.17 |
| 0% | 100% | Validation | 3.46 | 11.18 | 17.30 | 3.46 | 11.36 | 17.07 |
| 100% | 13% | Validation | 2.99 | 9.91 | 16.03 | 2.24 | 8.24 | 13.26 |
| 100% | 100% | Validation | 5.88 | 17.12 | 24.62 | 5.47 | 15.62 | 23.47 |

From Tab. 7, we can see that when fine-tuning with YDC training data of the same scale, the performance on the Youku-mPLUG validation set decreases due to the out-of-domain distribution. However, as the amount of YDC data increases, the generalization gains are significantly greater than the results obtained by training solely on Youku-mPLUG. This further demonstrates that the contribution of the proposed YDC data can enhance the performance of retrieval tasks.

## 7 CONCLUSION

In this paper, we address the significant gap in Chinese dense video captioning resources by introducing the Youku Dense Caption dataset, the first large-scale, high-quality Chinese dense video captioning dataset. Comprising 31,466 short videos annotated with 311,921 Chinese captions, it is the largest and most detailed publicly available dataset for fine-grained descriptions of Chinese video content. Extensive experiments and evaluations using state-of-the-art multi-modal models demonstrate the dataset's utility and impact on improving model performance. Additionally, our work shows that incorporating the Youku Dense Caption dataset as supplementary training data significantly enhances model performance. This dataset not only provides a critical resource for Chinese video-language processing research but also encourages further advancements in this field. We hope this work inspires more scholars to contribute to developing Chinese video-language processing technologies.

ACKNOWLEDGEMENT

This research is supported by National Natural Science Foundation of China (Grant No.62276154), Research Center for Computer Network (Shenzhen) Ministry of Education, the Natural Science Foundation of Guangdong Province (Grant No.2023A1515012914 and 440300241033100801770), Basic Research Fund of Shenzhen City (Grant No.JCYJ20210324120012033, JCYJ20240813112009013 and GJHZ20240218113603006), the Major Key Project of PCL for Experiments and Applications (PCL2023A09 and PCL2022A05), Alibaba Research Intern Program.

ETHICS STATEMENT

The Youku Dense Caption dataset primarily features concepts and language expressions that are relevant at the time of its creation. As language naturally evolves with human activities, our dataset may not include emerging concepts, new words, or contemporary expressions in the future. The authors will handle the long-term maintenance of the Youku Dense Caption dataset and the benchmarks evaluated in our paper. The dataset will be available on Modelscope, with Alibaba Cloud providing backend support for the download service. To ensure stability, we will regularly check the dataset URLs and promptly address any broken links. Our datasets are released under the Creative Commons Attribution-NonCommercial-ShareAlike 4.0 International Public License ("CC BY-NC-SA 4.0"), along with additional terms outlined in this document. Users must agree to this license when downloading or using the datasets from our website.

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

## A    ANNOTATION GUIDELINES

**Objective**    The goal is to create detailed and coherent descriptions for the main scenes presented in the provided video, including specific start and end timestamps for each description. The final annotation files are released in a video format.

**Description Structure**    Each description must adhere to the following structural components:

(1) Subject: Use general identifiers to describe subjects, avoiding specific names. Acceptable terms include "man", "woman", "young man", "boy", "child", "girl", "doll", "elderly person", and "groom". Avoid proper nouns or distinct identifiers, such as "Yang Mi", "Cangnan Mountain Immortal", or "gangster". (2) Description of the Subject: Include adjectives that pertain to the subject's color, size, state, facial expressions, and emotions. (3) Description of Actions (Optional): Describe the actions performed by the subject, indicating their nature and purpose. (4) Description of the Scene (Optional): Specify the type of location, such as "school", "plaza", "playground", or "shopping mall", while avoiding specific site identifiers like "Mingzhu elementary school" or "Beijing city".

**Annotation Requirements**    (1) Each description sentence must contain a minimum of ten words and consist of at least three sentences for a video. (2) Descriptions should provide specific details for each component (subject, action, and scene) and articulate events occurring in the video while maintaining grammatical and semantic coherence. (3) Common referential terms (e.g., "he", "she") may be used to reference previously introduced subjects. (4) A coherent narrative that encapsulates the primary content of the video should be produced, along with descriptions of noteworthy but non-primary scenes. (5) Annotations for overlapping video timestamps may be permissible. If multiple scenes are present in a single video, they should be described separately.

**Annotation Process** (1) Complete Viewing: View the entire video to grasp the overall content. (2) Drafting Descriptions: After viewing, create comprehensive descriptions based on the identified scenes. Each description should be coherent and reflect the content accurately. (3) Formatting Descriptions: Convert the drafted descriptions into the specified format: (description, [start time (s), end time (s)]) corresponding to the actual playback positions.

**Cleanup** Some segments of the video may contain nonsensical content that precludes valid descriptions, such as lengthy passages of text, meaningless advertisements, or "oral broadcasting" videos. In these instances, annotations for these clips should be omitted. Furthermore, we have excluded videos lacking explicit category tags, as these videos often exhibit ambiguous classifications or consist of composite footage from multiple genres. Additionally, issues are identified in the annotations, such as excessively brief annotation texts or an insufficient number of annotations per video. These instances of low-quality annotations, along with the associated videos, are also removed from the dataset.

**Timeline** The annotation process is expected to take approximately two months. This includes: (1) One round of initial marking by a single annotator. (2) A quality check conducted by a second annotator. (3) A final review to ensure completeness and accuracy.

By adhering to these guidelines, annotators ensure that the descriptions produced are both insightful and reflective of the video's content, enhancing its utility for review and analysis.

**Details** Our annotation team consisted of 1 annotation leader and 10 annotators. The leader was responsible for aligning annotation standards, conducting trial annotations, training annotators, and quality control. The annotators focused on specific annotation tasks and data revisions.

We engaged a professional data annotation agency. The annotation leader had at least 3 years of experience in managing annotation tasks, while the annotators had a minimum of 3 months of experience in CV-related data annotation. All team members were native Chinese speakers.

The annotation cost was 0.63 RMB per valid clip caption, with total video costs varying based on the number of valid slices. The entire process took 2 months, with efficiency improving over time as annotators became more proficient.

While we don't have detailed information about the internal labor distribution within the annotation agency, we understand from our interactions with the annotation leader that the workload was generally evenly distributed among annotators.

The annotation leader conducted quality checks on 10% of each batch of annotations. A 97% pass rate was required for batch approval; otherwise, the entire batch was returned for re-annotation. In the early stages, we also performed secondary reviews of the data to ensure alignment with annotation standards.

## B  DETAILED DATA STATISTICS

The specific number of categories and subcategories for the entire Youku Dense Caption dataset is shown in the Tab. 8. There are a total of 31,466 videos, categorized into 11 major categories and 84 subcategories.

When dividing benchmark data and non-benchmark data, we fixed the random seed to 42 and randomly selected 10% of the video IDs from the Youku Dense Caption dataset as the source for creating benchmark data. This subset consists of 3,185 videos and their corresponding 31,553 annotations. The remaining 90% of the data, comprising 28,281 videos and 280,368 annotations, is used as the source for training data.

During the quality filtering process for the retrieval benchmark, the text2video part was filtered from the original 31,553 texts down to 28,988 texts for retrieval evaluation, with an average of 4.96 related video IDs per text. For the video2text part, we simply converted the corresponding relationships. For the retrieval task's training data, we did not perform any quality filtering and used the original single text-video pairs to form the training data.

Table 8: Dataset Overview by Category and Subcategory

| Category | Number of Items | Category | Number of Items | Category | Number of Items |
|---|---|---|---|---|---|
| **Agriculture (三农)** | 2,518 | **Sports (体育)** | 2,722 | **Children (少儿)** | 2,006 |
| Agricultural Technology (农产家术) | 1,157 | Comprehensive Sports (体育综合) | 315 | Parenting and Childcare (亲子育儿) | 680 |
| Rural Cuisine (农村菜) | 973 | Sports News (体育资讯) | 67 | Early Childhood Education (少儿启蒙) | 165 |
| Hometown Specialties (家乡特色) | 144 | Fitness (健身) | 496 | Children's Talents (少儿才艺) | 1,082 |
| Agricultural Policies (惠农政策) | 244 | Extreme Sports (极限运动) | 123 | Comprehensive Children (少儿综合) | 18 |
|  |  | Competitive Sports (竞技体育) | 787 | Children's Variety Shows (少儿综艺) | 61 |
|  |  | Basketball (篮球) | 476 |  |  |
|  |  | Football (足球) | 458 |  |  |
| **Movies and TV (影视)** | 7,291 | **Comedy (搞笑)** | 1,861 | **Culture and Arts (文化艺术)** | 964 |
| Animation (动漫) | 49 | Comedy Theater (搞笑剧场) | 1,019 | Music Videos (MV) | 42 |
| Movie Clips (电影剪辑) | 2,023 | Comedy Skits (搞笑段子) | 769 | Instrumental Performances (乐器演奏) | 62 |
| Movie Extras (预告/杂谈) | 361 | Comprehensive Comedy (搞笑综合) | 73 | Traditional Culture (传统文化) | 195 |
| TV Show Clips (电视剧剪辑) | 2,320 |  |  | History (历史) | 95 |
| TV Show Extras (预告/杂谈) | 284 |  |  | Formal Education (学历教育) | 149 |
| Documentary Clips (纪录剪辑) | 2,134 |  |  | Family Education (家庭教育) | 54 |
| Documentary Extras | 120 |  |  | Literature and Arts (文学艺术) | 28 |
|  |  |  |  | Cross Talk and Sketches (相声小品) | 142 |
|  |  |  |  | Covers (翻唱) | 64 |
|  |  |  |  | Language Learning (语言学习) | 31 |
|  |  |  |  | Fine Arts (高雅艺术) | 102 |
| **Fashion and Entertainment (时尚娱乐)** | 2,411 | **Lifestyle (生活)** | 7,366 | **Science and Technology (科学技术)** | 1,769 |
| Runway Shows (T台走秀) | 471 | Hobbies (兴趣爱好) | 62 | Traditional Technology (传统科技) | 95 |
| Chinese Dance (中国舞) | 64 | Health and Wellness (养生保健) | 108 | Medical and Health (医疗健康) | 69 |
| Hair Styling (发艺) | 79 | Oddities (奇闻) | 180 | Digital New Media (数字新媒体) | 1,024 |
| Classical Dance (古典舞) | 72 | Pets (宠物) | 233 | Social Sciences (社会科学) | 46 |
| Ballroom Dance (国标舞) | 72 | Home Decor (家居装饰) | 81 | Science Popularization (科学科普) | 535 |
| Otaku Dance (宅舞) | 79 | Real Estate and Renovation (房产装修) | 63 |  |  |
| Square Dance (广场舞) | 60 | Travel (旅游) | 107 |  |  |
| Gaming (游戏) | 20 | Daily Life (日常生活) | 429 |  |  |
| Modern Dance (现代舞) | 63 | Astrology (星座运势) | 18 |  |  |
| Fashion (穿搭) | 556 | Automobiles (汽车) | 1,819 |  |  |
| Beauty (美妆) | 416 | Trendy Toys (潮玩) | 93 |  |  |
| Skincare (美容护肤) | 143 | Life Hacks (生活百科) | 275 |  |  |
| Nail Art (美甲) | 17 | Rural Cuisine (田园美食) | 209 |  |  |
| Comprehensive Dance (舞蹈综合) | 56 | Food (美食) | 1,992 |  |  |
| Street Dance (街舞) | 72 | Career and Workplace (职业职场) | 233 |  |  |
| Rap (说唱) | 29 | Cute Babies Daily (萌娃日常) | 1,372 |  |  |
| Live Music (音乐现场) | 75 | Street Interviews (街访) | 92 |  |  |
| Music News (音乐资讯) | 67 |  |  |  |  |
| **Variety Shows (综艺)** | 1,347 | **Finance (财经)** | 1,211 |  |  |
| Variety Show Clips (综艺剪辑) | 1,191 | Real Estate (房地产) | 49 |  |  |
| Variety Show Discussions (综艺杂谈) | 98 | Investment and Financial Management (投资理财) | 629 |  |  |
| Variety Show Previews and News (综艺预告资讯) | 58 | Finance and Business (财经商业) | 215 |  |  |
|  |  | Financial Experts (财经大咖) | 318 |  |  |

For the generation and localization tasks' benchmark quality filtering, we first applied a threshold filtering method based on the average self-BLEU value of annotated texts within a single video, reducing the original 3,185 videos to 2,746 videos. Then, by applying a threshold based on the color histogram correlation of video segments corresponding to timestamps, we further filtered down to 1,872 videos as the benchmark data, with an average of 10.7 text annotations per video. For the training data of the localization and generation tasks, we filtered the original training set to 16120 videos, with an average of 10.82 annotations per video.

Finally, to facilitate benchmark evaluation, we additionally selected 282 videos from the original unfiltered 3,185 videos' test data to create a mini test set, using the same post-processing method to produce the mini benchmark. The experimental results in this paper are all derived from the mini benchmark evaluations.

## C  PIPELINE OF BENCHMARK SETUP.

We randomly select 10% of the original dataset using a fixed seed to create the benchmark. These data annotations, if used as training data or for other data augmentation purposes, do not require post-processing due to their high quality and rich variety of categories. However, as benchmark data, they may contain some redundant data, leading to biased and less objective evaluations. Additionally, the annotation style or video type may not be suitable for certain tasks. For example, in temporal localization tasks, if a video's scene changes are minimal or the annotation texts are quite similar and lack distinguishing information, it can be difficult even for humans to differentiate. In fact, when given a video segment and its corresponding text annotation, the data relationship is reasonable. However, such data is only suitable for data augmentation and not for evaluation due to its characteristic of being reasonable in forward annotation but ambiguous in reverse inference. Therefore, we adopt the data post-processing method shown in the figure below to establish an objective and comprehensive benchmark.

## D  IMPLEMENTATION DETAILS OF BASELINES

In the initial data processing, the videos include both landscape and portrait orientations. We first resize them to either 360x640 or 640x360 dimensions, and standardize the frame rate to 15 FPS.

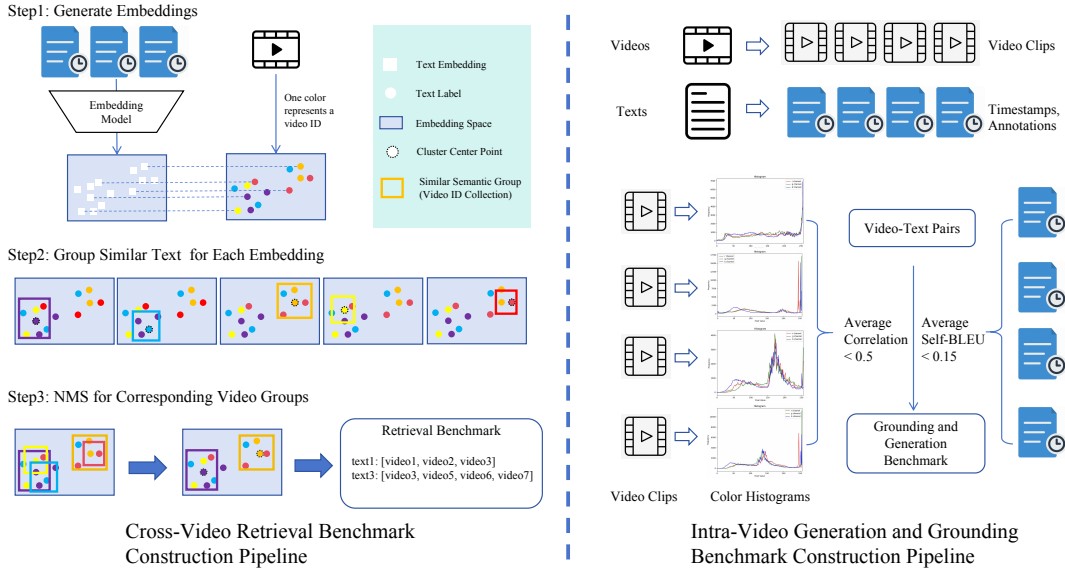

Figure 4: Pipeline of benchmark setup.

Then, using the processing script from the somethingsomething-v2 dataset, we convert the videos to a 320p format and remove the audio component.

**ViCLIP**   ViCLIP does not have many version distinctions; we use the default official demo code to uniformly extract 8 frames from the video for encoding through ViCLIP. For the localization benchmark evaluation, since moment-detr requires features for every two seconds of the video, we segment the video into two-second clips and then encode these segments using ViCLIP to serve as the features for these two-second clips.

**InternVideo2**   In the generation task, we use the InternVideo2-8B-Chat version, which consists of the visual encoding model from InternVideo2-1B and the language decoding model from Mistral-7B. For video processing, we uniformly extract 8 frames from the video as input, and the Chinese prompt input to the model is *Use a sentence to describe the video. Please answer in Chinese..* In the retrieval task, we use the InternVideo2-stage2_1b-224p-f4 model, which is trained in the second stage of InternVideo2. The input frame limit is 4, so we uniformly extract four frames from the video for encoding and retrieval.

**InternVL2**   In the generation benchmark evaluation, we use the official default demo for generation. We uniformly extract 8 frames from the video as video features and use the Chinese prompt *Use a sentence to describe the video.* for caption generation.

**CLIP**   For the grounding benchmark evaluation, we use the base-scale 512-dimensional CLIP and Chinese CLIP models. In the retrieval benchmark evaluation, we use the huge-scale 1024-dimensional models. Since CLIP does not have a huge version, we use the huge version of DFN-CLIP for our experiments. For feature extraction in the localization benchmark, we select the middle frame of each two-second video segment to encode and represent the features of that segment.

**mPLUG-Owl3**   As a multi-modal generative model, mPLUG-Owl3 first encodes visual data into embeddings, then inputs both the text prompt and visual embeddings into the language model simultaneously. In the retrieval benchmark evaluation, we use the visual embeddings encoded by the model as visual features, and input the text prompt into the language model to obtain the output hidden vectors. We then perform mean pooling on these hidden vectors to obtain the text features. Note that the number of frames extracted from the video is 16.

**Qwen2-VL** In the generation benchmark evaluation, Qwen2-VL dynamically extracts video frames, meaning the longer the video, the more frames are extracted. This is made possible by its unique architecture, which supports dynamic resolution input.

Table 9: Comparison of word frequency. We uniformly selected 20 words from the top 1000 high-frequency words before and after filtering for display.

| Rank | Original Annotation Word | Original Freq. | Processed Word | Processed Freq. |
|---|---|---|---|---|
| 25 | 一边 (One side) | 0.48611 | 黄色 (Yellow) | 0.45717 |
| 75 | 男士 (Gentleman) | 0.19027 | 西装 (Suit) | 0.18388 |
| 125 | 手指 (Finger) | 0.10943 | 狮子 (Lion) | 0.11035 |
| 175 | 西服 (Suit) | 0.07742 | 一把 (A handful) | 0.08013 |
| 225 | 外面 (Outside) | 0.06525 | 一台 (One unit) | 0.06621 |
| 275 | 房间内 (Inside the room) | 0.05288 | 美味 (Delicious) | 0.05445 |
| 325 | 黯黑 (Dark) | 0.04419 | 牛仔裤 (Jeans) | 0.04496 |
| 375 | 毛衣 (Sweater) | 0.03774 | 加入 (Join) | 0.03805 |
| 425 | 安静 (Quiet) | 0.03356 | 赛场 (Arena) | 0.03444 |
| 475 | 走来走去 (Walk back and forth) | 0.02899 | 长裙 (Long dress) | 0.03001 |
| 525 | 桌面上 (On the desk) | 0.02641 | 点头 (Nod) | 0.02681 |
| 575 | 墙边 (By the wall) | 0.02383 | 阳光 (Sunshine) | 0.02424 |
| 625 | 电动车 (Electric vehicle) | 0.02164 | 愤怒 (Angry) | 0.02248 |
| 675 | 撕咬 (Tear with teeth) | 0.02023 | 大厅 (Hall) | 0.02063 |
| 725 | 花白 (Grizzled) | 0.01881 | 花纹 (Pattern) | 0.01928 |
| 775 | 剪刀 (Scissors) | 0.01752 | 轮胎 (Tire) | 0.01836 |
| 825 | 工作服 (Work clothes) | 0.01642 | 笔直 (Straight) | 0.01702 |
| 875 | 开着车 (Driving) | 0.01546 | 慢悠悠 (Slowly) | 0.01588 |
| 925 | 卡其色 (Khaki) | 0.01443 | 头顶 (Top of head) | 0.01475 |
| 975 | 刷子 (Brush) | 0.01353 | 展厅 (Exhibition hall) | 0.01403 |

**MiniCPM-V** In the generation experiments with MiniCPM-V, we uniformly extract 64 frames from the video for generation. The Chinese prompt used is *Briefly describe this video in one sentence* because it always tends to generate overly long video descriptions.

**Training Details** In the data augmentation experiments for the generation task, we train the data for 1 epoch using the default fine-tuning parameters of the Swift framework. The learning rate is set to 1e-4, and the batch size is set to 16. The training is conducted on eight A800 GPUs, resulting in an effective batch size of 128.

In the data augmentation experiments for the retrieval task, we train the data for 10 epochs with a learning rate of 8e-4. The batch size per GPU is set to 256, and the training is conducted on eight A800 GPUs, resulting in an effective batch size of 2048.

# E  VOCABULARY STATISTICS

Note that in Section 4, we employed methods for data filtering in downstream tasks. Here, we will compare the vocabulary before and after filtering, analyzing the changes that occurred.

From Table 9, it can be seen that before processing, the frequency of high-frequency words is higher, indicating a higher redundancy in the annotations. After processing, the frequency of high-frequency words decreases, while the frequency of low-frequency words slightly increases, suggesting that the overall frequency distribution of the word list becomes more balanced. This can also be observed from the range and standard deviation shown in Table 10, where the range and standard deviation of word list frequencies in the processed data are smaller.

Table 10: Comparison of the range and standard variance before and after processing.

| Indicator | Original Annotation | Processed Data |
|---|---|---|
| Range of Word Frequencies | 0.4726 | 0.4431 |
| Standard Deviation of Word Frequencies | 0.1928 | 0.1824 |

## F ABLATION ANALYSIS ABOUT SELF-BLEU

Regarding the self-BLEU threshold of 0.15 adopted in the benchmark setup, we will discuss some examples here.

Table 11: Some examples with self-BLEU values between 0.15-0.17, where words highlighted in red or in orange indicate higher semantic repetition. It can be observed that while the overall semantics continuously change, certain parts of the meaning remain relatively similar.

| Timestamps | Captions | Timestamps | Captions |
|---|---|---|---|
| 0.25-12.76 | 一个有着卷发的女人在舞台上唱歌
A curly-haired woman singing on stage | 5.77-10.70 | 黑色西装的年轻男子们正在礼貌的打招呼
Young men in black suits are politely greeting each other |
| 13.38-25.27 | 一个长头发女人在舞台上一边走着一边唱着歌
A long-haired woman walking and singing on stage | 14.88-18.23 | 穿着黑色西装的年轻男子们正聚集在一起讲台
Young men in black suits are gathered together talking |
| 26.26-28.11 | 一个女人在舞台上扭头看着左边的另一个女人唱歌
A woman on stage turning her head to look at another woman singing on the left | 18.23-23.92 | 穿着黑色衣服的中年男子们走进了室内
Middle-aged men in black clothes are walking indoors |
| 28.73-32.82 | 在舞台上有两个女人，其中一个有着卷发的女人在唱歌
Two women on stage, one with curly hair is singing | 40.01-46.82 | 西装的中年男子们正在互相打招呼
Middle-aged men in suits are greeting each other |
| 33.69-39.49 | 一个长头发的女人在舞台上拿着话筒唱歌
A long-haired woman singing with a microphone on stage | 46.82-55.31 | 一群人聚集在一起聊天手里还拿着酒杯
A group of people gathered together chatting with drinks in hand |
| 39.99-43.11 | 在舞台上两个拿着话筒的女人在唱歌
Two women singing with microphones on stage | 55.31-65.42 | 端着酒杯的中年男子们正在握手交流
Middle-aged men holding drinks are shaking hands and communicating |
| 43.60-45.97 | 一个戴着耳坠的女人高兴的唱着歌
A woman with earrings happily singing | 76.27-83.56 | 戴着眼镜的中年男子正在跟旁边的人说话
A middle-aged man wearing glasses is talking to the person next to him |
| 46.59-52.58 | 一个长头发的女人左右歪着头唱歌
A long-haired woman singing while tilting her head left and right | 83.56-88.11 | 黑色衣服的中年男子正在边走路边和旁边的人聊天
A middle-aged man in black clothes is walking and chatting with the person next to him |
| 0.00-10.65 | 两位穿着黑色衣服的主持人坐在桌子旁边说话
Two hosts in black clothes sit at a table talking | 0.00-6.94 | 一名穿着黑色西装的男孩子在镜头前展示
A boy in a black suit is presenting in front of the camera |
| 12.29-16.23 | 一群篮球运动员在篮球场上奔跑着打篮球
A group of basketball players are running and playing on the court | 6.94-14.88 | 剪着短发的男孩子在镜头前说着话
A boy with short hair is talking in front of the camera |
| 16.74-19.21 | 穿着白色衣服的运动员在抢夺着黑色衣服运动员手中的篮球
Athletes in white clothes are trying to steal the ball from athletes in black clothes | 14.88-23.63 | 穿着黄色的鞋黑色的裤子的男孩在镜头前表演着节目
A boy wearing yellow shoes and black pants is performing in front of the camera |
| 22.24-28.68 | 穿蓝色衣服的人和穿白色衣服的人在打篮球
People in blue clothes and white clothes are playing basketball | 23.63-34.91 | 在粉色的幕布前面站着一个小男孩在说话
A little boy is standing in front of a pink curtain, talking |
| 31.27-40.34 | 篮球场上穿着红色衣服的人和穿着白色衣服的人在打篮球
People in red and white clothes are playing basketball on the court | 34.91-46.06 | 黄色的木制地板上站着一个小男孩在说着话
A little boy is standing on a yellow wooden floor, talking |
| 43.20-51.20 | 穿着蓝色衣服的运动员在穿着白色衣服的运动员的围堵下把球投入了球框中
The athlete in blue clothes shoots the ball into the hoop while surrounded by athletes in white clothes | 46.06-61.85 | 衣着考究的男孩子在镜头前面毫不怯场
A well-dressed boy appears confident in front of the camera |
| 54.26-67.07 | 穿着白色球衣的运动员把篮球投入了球框中
The athlete in a white jersey shoots the basketball into the hoop | 61.85-76.44 | 说话落落大方得体的男孩子在镜头前展示着
A boy speaks eloquently and appropriately while presenting in front of the camera |

It's worth noting that for samples far from the 0.15 threshold, such as those with self-BLEU values of 0.8 or 0.01, these are extreme cases. Obviously, samples with high self-BLEU values have lower textual diversity, while samples with low self-BLEU values have higher textual diversity. Therefore, to discuss the threshold setting, we will only observe the diversity differences in samples with self-BLEU values around 0.15, as shown in the Table 11 and Table 12.

From Table 11, we can observe that in samples where the self-BLEU value is slightly above 0.15, there is almost always a certain recurring phrase that appears in all sentences, along with some repeated phrases occurring in most sentences, such as "在舞台上唱歌" (singing on stage) or "打招呼" (greeting) or "在镜头前" (in front of the camera), or there are sentences that are semantically almost identical, like (a group of basketball players playing basketball), etc. From Table 12, we can see that repeated phrases only appear in part of the sentences, for example, "弹钢琴" (playing the piano) and "穿着西装带着眼镜的男人" (a man wearing a suit and glasses) appear in 5 out of 7 instances in our given samples, and there is almost no additional repetition in content. In most samples, repeated phrases like "穿西装的男人" (man in a suit), "聊天" (chatting), or "说话" (talking) only appear in a few sentences. Looking at the overall description changes, we can infer that setting the self-BLEU to 0.15 can effectively prevent certain phrases from appearing too frequently in the overall annotation, thereby ensuring the diversity of the annotated text.

## G CHINESE CHARACTERISTICS

For instance, admiring the full moon during the Mid-Autumn Festival is an important activity in traditional Chinese culture. The Mid-Autumn Festival, also known as the Mooncake Festival, is celebrated on the 15th day of the eighth month of the lunar calendar. On this day, people gather to

Table 12: Some examples with self-BLEU values between 0.13-0.15, where words highlighted in the same color indicate higher semantic repetition. It can be observed that even the most frequently occurring words appear in only part of the sentences, and there is an increasing diversity of repeated words appearing in these sentences.

| Timestamps | Captions | Timestamps | Captions |
|---|---|---|---|
| 0.00-14.72 | 一个穿着西装的男人和一个穿着围裙的男人在聊天
A man in a suit and a man in an apron are chatting | 0.00-13.33 | 一个跨着包包的女子站在舞台上说话
A woman with a bag is speaking on stage |
| 14.72-29.48 | 穿着围裙的男人匍匐在地上做着各种动作
The man in an apron is crawling on the ground making various movements | 13.33-15.88 | 长相帅气的男子坐在那里哈哈笑着
A handsome man is sitting there laughing |
| 29.48-38.69 | 穿着西装的男人指着趴在地上的男子在说话
The man in a suit is pointing at the man on the ground and speaking | 15.88-28.07 | 穿着白色蕾丝衣服的女子站在舞台上说话
A woman in white lace clothing is speaking on stage |
| 38.69-51.49 | 趴在地上的男子在地上做着各种动作
The man lying on the ground is making various movements | 28.07-30.06 | 穿着蓝色衣服的女子坐在那里看着
A woman in blue clothing is sitting there watching |
| 51.49-61.26 | 站着的男子指着坐在地上的男子说着话
The standing man is pointing at the man sitting on the ground and speaking | 30.06-35.72 | 一个戴着项链的男子坐在那里说话
A man wearing a necklace is sitting there speaking |
| 61.26-69.49 | 穿西装的男子坐在椅子上拍了一下桌子
The man in a suit sits in a chair and slaps the table | 35.72-44.24 | 女子站在那里摆弄着手上的黑色包包
A woman is standing there fiddling with a black bag in her hands |
| 69.49-84.87 | 坐在地上的男子站起来和坐在椅子上的男人说着话
The man sitting on the ground stands up and talks to the man sitting in the chair | 44.24-46.62 | 一个女和一个男子坐在那里大笑着
A woman and a man are sitting there laughing loudly |
| 84.87-97.14 | 坐在椅子上的男人拿了一根筷子在吃
The man sitting in the chair takes a chopstick and is eating | 46.62-53.19 | 戴着耳环的女子拿着包包站在那里说话
A woman wearing earrings is standing there with a bag, speaking |
| 97.14-101.67 | 穿着粉色T恤的男人匍匐在地上说着话
A man in a pink T-shirt is crawling on the ground speaking | 53.19-58.82 | 女子把手里的包包递给了穿着蓝色衣服的女子
The woman hands the bag to the woman in blue clothing |
| 0.32-15.49 | 一个秃头的男子正在弹奏着黑色钢琴
A bald man is playing a black piano | 3.46-7.95 | 穿着西服戴着眼镜的男人在墙上画画
A man in a suit and glasses is drawing on the wall |
| 15.49-36.17 | 一个男子正在盖着绿色布匹的钢琴前坐着
A man is sitting in front of a piano covered with a green cloth | 7.95-13.04 | 身穿西服戴着眼镜的男人在教几位小朋友画画
A man in a suit and glasses is teaching several children how to draw |
| 36.17-55.52 | 一个戴着眼镜的男子正在看着前方
A man wearing glasses is looking ahead | 13.04-19.55 | 戴着眼镜的男人在讲解墙壁上的画作
The man with glasses is explaining the artwork on the wall |
| 55.52-69.41 | 一个穿着黑色衬衣的男子正在弹奏着钢琴
A man in a black shirt is playing the piano | 21.23-24.38 | 几位小朋友围在身穿西服戴着眼镜的男人边上看他画画
Several children gather around the man in a suit and glasses to watch him draw |
| 69.41-83.29 | 一双纤细的手正在弹奏着黑色钢琴
A pair of slender hands are playing a black piano | 24.38-36.94 | 许多小朋友在一张长桌上学习着画画的技巧
Many children are learning drawing techniques at a long table |
| 83.29-96.67 | 一个男子正笔直的坐在钢琴前弹奏着
A man is sitting straight in front of the piano, playing | 36.94-39.89 | 一位身穿棉服戴着口罩的小朋友趴在桌子上画着画
A child wearing a cotton jacket and mask is lying on the table drawing |
| 96.67-117.17 | 一个男子一脸严肃的看着前方弹奏着钢琴
A man is playing the piano with a serious expression, looking ahead | 49.59-52.22 | 拿着毛笔的小朋友围在身穿西服戴着眼镜的男人边上看他画画
Children holding brushes gather around the man in a suit and glasses to watch him draw |

eat mooncakes and admire the full moon, expressing their wishes for family reunion. This cultural practice frequently appears in Chinese videos and annotations but is seldom seen in English datasets. For example, in the Youku Dense Caption dataset, there are videos describing the process of making mooncakes and of people gathering to eat them.

Similarly, the tradition of paying New Year visits is an integral part of Chinese culture. The Chinese New Year is the most important traditional festival in China, marking the beginning of the lunar new year. During this period, paying visits to family and friends to offer blessings and express hopes for a prosperous new year is a significant activity. This custom holds a prominent place in Chinese datasets, with specific practices of New Year visits and celebrations appearing in the Youku Dense Caption dataset. In contrast, such details are often overlooked, simplified, or nearly absent from English datasets, where similar New Year video content is rarely found.

Wrapping Zongzi on the Dragon Boat Festival is also one of the important traditional customs in China. This activity is usually accompanied by family reunion and ancestor worship, which has profound cultural significance and historical origins. For example, the sample of "1108754691" in Figure 3 describes the scene of a family gathering for dinner while the elders are cutting Zongzi and preparing to eat Zongzi. Similarly, the Chinese art cross-talk is unique in the Chinese context, with no equivalent in English. These cultural differences manifest not only in the distribution of textual annotations but also in video content.

# H MORE DETAILS

Additionally, we examined the issue of average annotation length, where a text annotation length greater than 10 is our minimum requirement. We present a comparison of annotation lengths from some existing manually annotated video caption datasets. As the first Chinese dense caption dataset, Youku Dense Caption not only has the highest number of annotations among manually annotated datasets but also boasts one of the longest average sentence lengths. Other video caption datasets with longer average sentence lengths, such as HD-VILA-100M with an average sentence length of 32.5 words, achieve this by using Automatic Speech Recognition (ASR) for annotation. It is worth noting that in most cases, ASR cannot accurately describe the visual content and may include a large number of colloquial and irrelevant words.

Table 13: Comparison of Annotation Quantity and Length with Other Manually Labeled Datasets.

| Dataset | Total Annotations | Average Annotation Length |
|---|---|---|
| MSR-VTT | 10k | 9.3 |
| LSMDC | 118k | 7.0 |
| YouCook2 | 14k | 8.8 |
| How2 | 80k | 20.0 |
| ANet caption | 100k | 13.5 |
| Youku Dense Caption | 311k | 17.9 |

# I MORE CASES IN YOUKU DENSE CAPTION

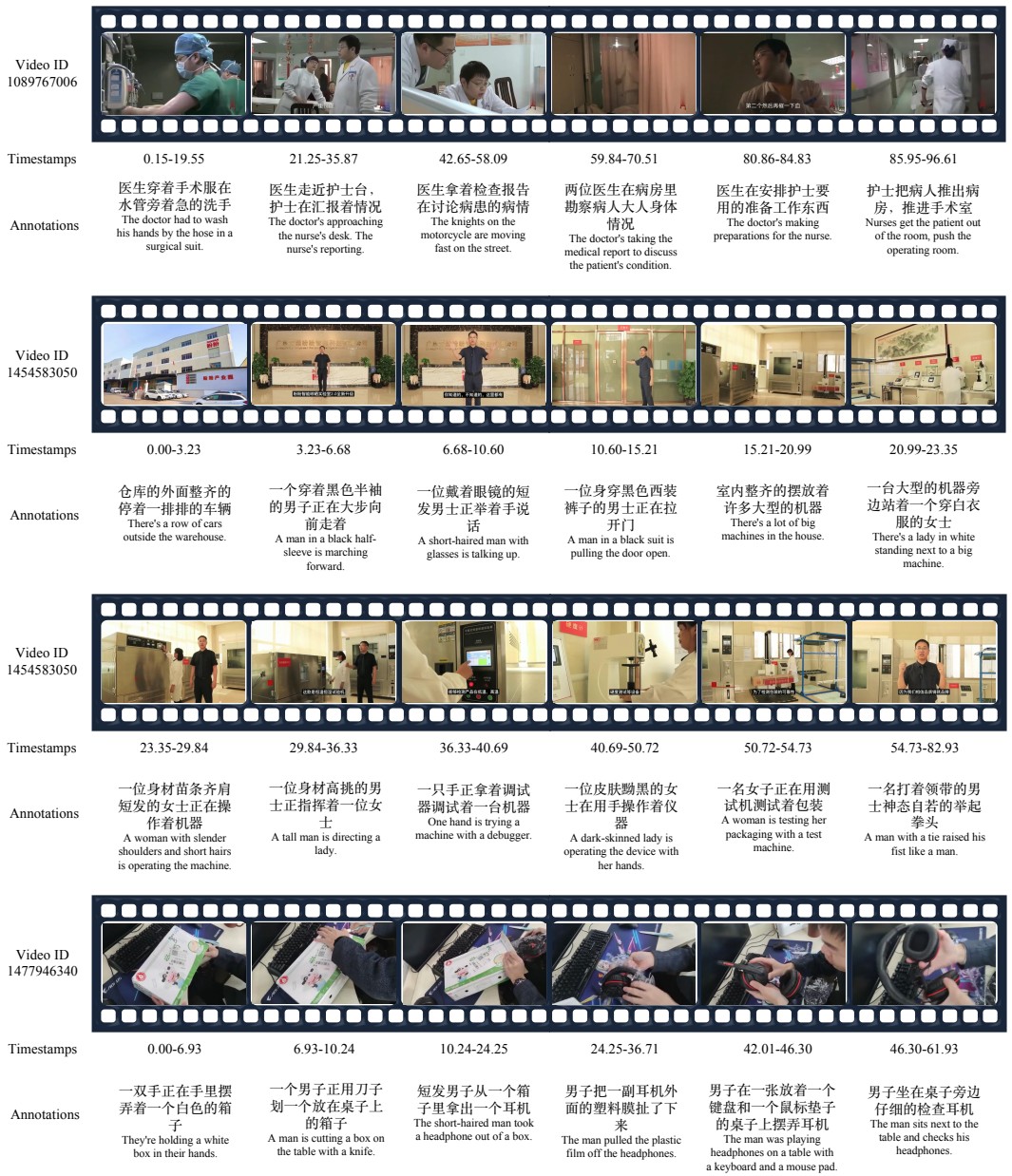

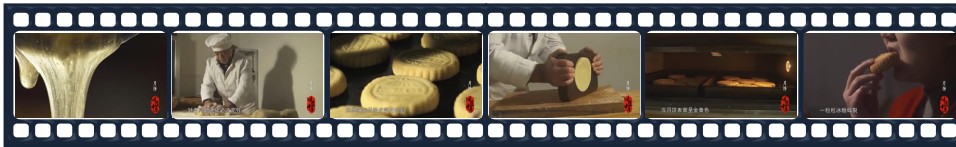

| | | | | | | |
|---|---|---|---|---|---|---|
| Video ID 1473367532 | | | | | | |
| Timestamps | 0.00-4.00 | 4.00-13.73 | 13.73-19.86 | 19.86-30.40 | 30.40-58.52 | 63.99-86.79 |
| Annotations | 晶莹剔透的糖制品看着让人垂涎欲滴 The crystal-crysted sugar products look like they're salivating. | 穿着白色大褂带着白色帽子的大爷站在中间 He's standing in the middle of a big white coat with a white hat. | 一块块带着文字和花纹的月饼被制作出来 A mooncake with words and strips was made. | 面团经过两个人的揉搓然后放进精致的模具中加工 The face passes through the twigs of two men and is processed in fine molds. | 月饼经过专业烤箱一段时间的高温烘焙逐渐成型 The mooncakes have evolved through a period of hot baking in a professional oven. | 男士拿起月饼咬了一口满嘴流露出芳香 The man took the mooncake and bit it with his mouth full of fragrance. |

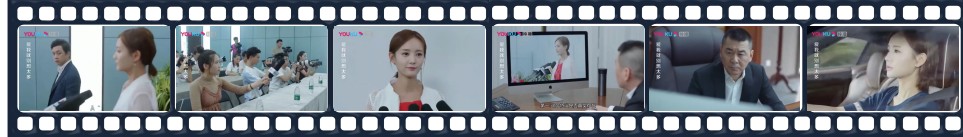

| | | | | | | |
|---|---|---|---|---|---|---|
| Video ID 1216057735 | | | | | | |
| Timestamps | 0.00-1.00 | 8.86-10.41 | 22.71-32.44 | 32.44-37.22 | 50.03-51.59 | 56.97-62.96 |
| Annotations | 一位美女走上台讲话台下男子疑惑的看着她 A beautiful woman comes up on stage and looks at her, and men wonder about her. | 台下的记者又的举着摄像机有的拿着话筒 The reporter on the stage held the phone with the camera. | 女子还在台上讲着关于自己的事情 The woman's still talking about herself on the stage. | 坐在桌前的男子手里握着鼠标动情的看着屏幕中的女子 The man sitting at the table looking at the woman on the screen with the mouse in his hand. | 男子坐在电脑前开始流露出不可思议的表情 A man sitting in front of a computer starts to show his incredible face. | 女子开着一辆红色的轿车飞驰在公路上 The woman was riding on the road in a red sedan. |

