# OpenReview forum: "Youku Dense Caption: A Large-scale Chinese Video Dense Caption Dataset and Benchmarks"
_ICLR.cc/2025/Conference — ICLR 2025 Poster_

### Official Review · Reviewer_Qgjf · 2024-10-20

**Soundness:** 2
**Presentation:** 2
**Contribution:** 3
**Rating:** 6
**Confidence:** 5

**Summary:**

The paper introduces Youku Dense Caption dataset, which is the largest publicly available Chinese dense video captioning dataset for now. The dataset is annotated by human to guarantee the quality of the dataset.
Building upon the proposed dataset, the paper also establishes benchmarks for video-language tasks.
The experiments demonstrate that existing state-of-the-art multi-modal models can benefit from this dataset.

**Strengths:**

- The paper focuses on the topic of Chinese dense video captioning dataset, which is an interesting and under-explored research area.

**Weaknesses:**

- Doubtful value of the proposed dataset
    - First, while the proposed dataset is claimed to be a dense video captioning dataset, the collection pipeline is similar to regular video captioning datasets, like HD-VILA-100M or Panda-70M, where a long video is first segmented into multiple clips and each clip is annotated with a caption. Could the authors provide more differences between the dataset collection pipelines of your dataset and a regular video-text dataset?
    - Second, it is unconvincing to state that "Chinese and English have significant linguistic differences, so a Chinese dataset is needed". I appreciate the authors show the errors of translation in line 211 and Section 3.2.2. However, I use ChatGPT to translate the provided samples and it can produce correct results in most of the cases. Take the leftmost sample in Figure 3 as example, I got this: "A group of motorcyclists is resting by the roadside, chatting." from ChatGPT, which is totally correct.

- Lack of necessary experiments: to evaluate the value of the proposed dataset, the authors need to train a model on different datasets and show that the one trained on the proposed dataset is more robust than the others. Such experiment should be conducted on different tasks, such as dense video generation, partially relevant video retrieval. However, none of this experiment is presented.

- It has been shown that long and detailed prompts are beneficial to various tasks, such as video generation. However, the caption annotations are short and less detailed, limiting the value of the dataset.

- For the scene change detection algorithm mentioned in lines 345~364, TransNet-v2 should be more robust than the adopted pixel-based algorithm.

**Questions:**

- How is a long video segmented into multiple clips? What is the splitting criteria?

- Could you show more data samples in Youku Dense Caption dataset? It is helpful for checking the diversity, visual quality, and annotation quality  of the proposed dataset?

---

> ### Author Response · Authors · 2024-11-15
> **Rebuttal by Authors**
>
> Thank you for your comprehensive review and insightful questions. We greatly appreciate your feedback and would like to address your concerns as follows:
>
> Regarding Weakness 1.1:
>
> Unlike datasets such as HD-VILA-100M or Panda-70M that use ASR or image captioning models for annotation, our dataset is entirely manually annotated. Human annotators describe the videos and select informative segments for segmentation. This approach ensures that our annotations align with natural human language usage and undergo manual screening to maintain high quality.
>
> Regarding Weakness 1.2:
>
> While advanced language models like ChatGPT could indeed improve Chinese translation quality, our dataset's primary motivation extends beyond language differences to encompass cultural and content disparities between Chinese and foreign contexts. As mentioned in Section 3.2.2, unique Chinese cultural elements like tea ceremonies, traditional festivals (Enjoy the moon on Mid-Autumn Festival and pay New Year's greetings on New Year's Day), etc., are rarely found in public English datasets but are common in Chinese contexts. This difference is reflected not only in annotation text distribution but also in video content distribution. As noted by widely-used Chinese CLIP[1], bridging the gap caused by cultural differences in annotation and video content is our dataset's core value.
>
> Regarding Weakness 2:
>
> In Section 6, we demonstrate the effectiveness of our dataset by training Qwen2-VL-2B on the VATEX[2] Chinese dataset for dense video captioning and ViCLIP on the Youku-mPLUG[3] Chinese dataset for video retrieval. The results show improved performance on their own benchmarks, validating the utility of our dataset.
>
> Regarding Weakness 3:
>
> As mentioned in Appendix A, we've implemented strict requirements for annotation length and quantity. All annotation texts must exceed 10 characters, and each video must have more than 3 annotation texts, ensuring sufficient detail and comprehensiveness.
>
> Regarding Weakness 4:
>
> Appendix A details our annotation process. Video segment division is entirely manual. The color histogram correlation-based filtering algorithm mentioned in Section 4.2 is an additional step to enhance dataset quality by removing potentially similar segments from the manual divisions, thereby ensuring diversity.
>
> Regarding Question 1:
>
> Appendix A provides a comprehensive explanation of our annotation process. After detailed human descriptions of complete videos, annotators manually divide the annotation text and video segments. The division criteria are primarily based on human judgment of information content in video segments.
>
> Regarding Question 2:
>
> In the submitted supplemental materials, we've included all detailed annotation texts for generation, retrieval, and grounding tasks. For the retrieval benchmark, we've already divided and screened similar texts, ensuring that highly similar texts do not repeatedly appear in the dataset.
>
> We hope these responses address your concerns adequately. If you have any further questions or require additional clarification, please don't hesitate to ask. We appreciate your valuable feedback and are committed to improving our research based on your insights.
>
> [1] Yang, An, et al. "Chinese clip: Contrastive vision-language pretraining in chinese." arXiv preprint arXiv:2211.01335 (2022).
>
> [2] Wang, Xin, et al. "Vatex: A large-scale, high-quality multilingual dataset for video-and-language research." Proceedings of the IEEE/CVF international conference on computer vision. 2019.
>
> [3] Xu, Haiyang, et al. "Youku-mplug: A 10 million large-scale chinese video-language dataset for pre-training and benchmarks." arXiv preprint arXiv:2306.04362 (2023).

---

> > ### Comment · Reviewer_Qgjf · 2024-11-18
> > **Official Comment by Reviewer Qgjf**
> >
> > Thanks the authors for a prompt reply! I appreciate your effort. However, I still have some unfixed concerns or questions.
> >
> > For Weakness 1.1: my concern is not about how you annotate the caption labels. My question is why you categorize your dataset as a dense video dataset. HD-VILA-100M and Panda-70M collect video samples by segmenting long videos, which is also the way you collected Youku Dense Caption. What is the fundamental difference between Youku Dense Caption and them to make you call it a dense video dataset. For me, a dense video dataset should include temporal related annotation, such as the event-based captions for a consecutive video or the subject information within a video (for example: *A* first does something. *A* then do another thing with *B* joining him). You can check some examples in the webpage of ActivityNet (https://cs.stanford.edu/people/ranjaykrishna/densevid/).
> >
> > For Weakness 2: thanks for correcting me, Table 5 and 6 are actually the experiments I would like to see. However, it seems like data distribution dominates the results and makes the results less informative. For example, given the same scale of the training dataset (upper part of Table 5), using 100% VATEX and 0% Youku Dense Caption can perform the best on the validation set of VATEX, vice versa. While it is reasonable, could you use the other dataset as the testing dataset (e.g., Youku-mPLUG), so it could possibly avoid the dataset distribution issue? Another suggestion is: use "Captioning Performance" instead of "Generation Performance" which usually means image/video generation.
> >
> > For Weakness 3: > 10 characters is quite normal for existing large-scale video-captioning datasets. Given this scale of dataset, this captioning length is too short to provide valuable contribution for the community.

---

> > > ### Author Response · Authors · 2024-11-22
> > > **Rebuttal by Authors**
> > >
> > > Regarding weakness 1.1, Section 4 of the ActivityNet Captions paper states: "The ActivityNet Captions dataset connects videos to a series of temporally annotated sentences. Each sentence covers a unique segment of the video, describing an event that occurs." Our dataset follows the same principle.
> > >
> > > Dense video caption datasets provide multiple time-stamped descriptions for short segments throughout a video, offering detailed, temporally-localized annotations of specific events and actions. In contrast, traditional video caption datasets typically offer a single, high-level description for the entire video. This fundamental difference results in dense caption datasets having greater annotation density, more precise temporal information, higher level of detail, increased data volume, and greater complexity in both creation and associated tasks. While more challenging to produce and utilize, dense caption datasets enable more fine-grained video understanding and have advantages in applications requiring detailed content analysis and retrieval.
> > >
> > > And we also provide some examples from our Youku Dense Caption dataset and compared them with samples from the ActivityNet captions dataset, as shown below. It is evident that our dataset is highly similar to the ActivityNet caption dataset in both form and style.
> > >
> > > Regarding weakness 2, we appreciate your correction on the term "Captioning Performance". We have also provided additional test results on the Youku-mPLUG validation set, utilizing Vatex and Youku Dense Caption for training:
> > >
> > > | Training Data | BLEU-4 | METEOR | ROUGE-L | CIDEr |
> > > |---------------|--------|--------|---------|-------|
> > > | vatex         | 3.08     | 12.77     | 19.53      | 29.53    |
> > > | Youku Dense Caption          | 3.46     | 13.54     | 22.76      | 31.55    |
> > >
> > > Addressing weakness 3, regarding the issue of short average annotation length, a text annotation length greater than 10 is merely our minimum requirement. We present a comparison of annotation lengths from some existing manually annotated video caption datasets. As the first Chinese dense caption dataset, Youku Dense Caption not only has the highest number of annotations among manually annotated datasets but also boasts one of the longest average sentence lengths. Other video caption datasets with longer average sentence lengths, such as HD-VILA-100M with an average sentence length of 32.5 words, achieve this by using Automatic Speech Recognition (ASR) for annotation. It's worth noting that in most cases, ASR cannot accurately describe the visual content and may include a large number of colloquial and irrelevant words.
> > >
> > > | Dataset       | Total Annotations | Average Annotation Length |
> > > |---------------|-----------------|-------------------------|
> > > | MSR-VTT[1]       | 10k             | 9.3                     |
> > > | LSMDC[2]         | 118k            | 7.0                     |
> > > | YouCook2[3]      | 14k             | 8.8                     |
> > > | How2[4]          | 80k             | 20.0                    |
> > > | ANet caption[5]  | 100k            | 13.5                    |
> > > | Youku Dense Caption  | 311k     | 17.9                    |
> > >
> > > [1] Xu, Jun, et al. "Msr-vtt: A large video description dataset for bridging video and language." Proceedings of the IEEE conference on computer vision and pattern recognition. 2016.
> > >
> > > [2] Rohrbach, Anna, et al. "Movie description." International Journal of Computer Vision 123 (2017): 94-120.
> > >
> > > [3] Zhou L, Xu C, Corso J. Towards automatic learning of procedures from web instructional videos[C]//Proceedings of the AAAI Conference on Artificial Intelligence. 2018, 32(1).
> > >
> > > [4] Sanabria R, Caglayan O, Palaskar S, et al. How2: a large-scale dataset for multimodal language understanding[J]. arXiv preprint arXiv:1811.00347, 2018.
> > >
> > > [5] Krishna R, Hata K, Ren F, et al. Dense-captioning events in videos[C]//Proceedings of the IEEE international conference on computer vision. 2017: 706-715.

---

> > > > ### Author Response · Authors · 2024-11-22
> > > > **Rebuttal by Authors (continued)**
> > > >
> > > > We provide additional examples from the ActivityNet Caption dataset and ours, along with extra samples in Appendix F and Appendix G.
> > > >
> > > > **Youku Dense Caption:**
> > > >
> > > > "1478912125": {
> > > >     "7.25-14.01": "一位漂亮的美女在山里驻足抹汗",
> > > >     "16.86-22.36": "这位女子在泥土挖出一个东西",
> > > >     "24.13-29.25": "美女在河边把挖出的东西洗干净了并露出了笑容",
> > > >     "29.16-34.01": "穿着白色衣服的美女背着竹篓在奔跑",
> > > >     "34.14-38.20": "美女背着竹篓走过了一座古老拱桥",
> > > >     "38.40-40.83": "美女把竹篓交给一个穿黑衣服的老婆婆",
> > > >     "40.89-46.33": "美女拿起一个大酒罐给一个老爷爷倒酒喝",
> > > >     "55.10-57.76": "穿着枚红色旗袍的一群女子端着盘子走来",
> > > >     "57.91-59.23": "白发老爷爷一家人在吃饭其中一位女子帮他端了一碗酒"
> > > > }
> > > >
> > > > (English translation:
> > > > "7.25-14.01": "A beautiful woman stops to wipe sweat in the mountains",
> > > > "16.86-22.36": "The woman digs something out of the soil",
> > > > "24.13-29.25": "The beauty washes the excavated object clean by the river and smiles",
> > > > "29.16-34.01": "The beautiful woman in white clothes runs with a bamboo basket on her back",
> > > > "34.14-38.20": "The beauty walks across an ancient arched bridge with the bamboo basket",
> > > > "38.40-40.83": "The beauty hands the bamboo basket to an old lady in black clothes",
> > > > "40.89-46.33": "The beauty picks up a large wine jug and pours wine for an old man to drink",
> > > > "55.10-57.76": "A group of women in red cheongsams walk in carrying plates",
> > > > "57.91-59.23": "A white-haired old man and his family are eating, and one woman helps him with a bowl of wine")
> > > >
> > > > "1043362718": {
> > > >     "0.27-6.00": "视频中向我们展示了大自然中一群可爱的动物",
> > > >     "6.00-14.47": "一只黄鼠狼从一块木头后面慢慢的探出了头",
> > > >     "14.47-20.46": "然后这只黄鼠狼警惕的看着周围的环境",
> > > >     "20.46-24.66": "这只黄鼠狼走着走着就会站起来看着周围",
> > > >     "24.66-38.52": "一只黄鼠狼和一只松鼠正在草地上打架",
> > > >     "38.52-53.52": "随后黄鼠狼正在对着松鼠紧追不舍",
> > > >     "53.52-61.23": "最中黄鼠狼将这只小松鼠逮捕住了",
> > > >     "61.23-82.00": "然后黄鼠狼和松鼠便在草地上展开了斗争",
> > > >     "82.00-90.74": "视频向我们展示了大自然中缺一不可的自然生态"
> > > > }
> > > >
> > > > (English translation:
> > > > "0.27-6.00": "The video shows us a group of adorable animals in nature",
> > > > "6.00-14.47": "A weasel slowly pokes its head out from behind a piece of wood",
> > > > "14.47-20.46": "Then this weasel vigilantly looks at the surrounding environment",
> > > > "20.46-24.66": "This weasel walks and occasionally stands up to look around",
> > > > "24.66-38.52": "A weasel and a squirrel are fighting on the grass",
> > > > "38.52-53.52": "Subsequently, the weasel is chasing the squirrel relentlessly",
> > > > "53.52-61.23": "Finally, the weasel captures the small squirrel",
> > > > "61.23-82.00": "Then the weasel and squirrel engage in a struggle on the grass",
> > > > "82.00-90.74": "The video shows us the indispensable natural ecology in nature")
> > > >
> > > > "1290018961": {
> > > >     "0.00-2.28": "戴着黑色帽子的男子在户外烤着烤串",
> > > >     "2.28-5.83": "帅气的男子坐在凳子上品尝着锅里的美食",
> > > >     "5.83-11.21": "视频中有两个小孩子在开心的玩耍着",
> > > >     "11.21-16.15": "两个男子蹲在地上在鼓捣着什么东西",
> > > >     "16.15-17.66": "锅里面放着很多的美食，看着还是生的，要煮熟吃",
> > > >     "17.66-31.28": "男子坐在一个炉子旁边在烧着火，旁边小孩也在帮忙",
> > > >     "31.28-37.49": "两个小女孩开心的吃着手里的西瓜",
> > > >     "37.49-64.08": "几个小孩坐在墙头上欣赏着外面的风景",
> > > >     "64.08-76.50": "他们吃完饭就开着各自的车子离开了"
> > > > }
> > > >
> > > > (English translation:
> > > > "0.00-2.28": "A man wearing a black hat is grilling skewers outdoors",
> > > > "2.28-5.83": "A handsome man sits on a stool tasting the delicacies in the pot",
> > > > "5.83-11.21": "There are two children happily playing in the video",
> > > > "11.21-16.15": "Two men are squatting on the ground fiddling with something",
> > > > "16.15-17.66": "There's a lot of food in the pot, it looks raw and needs to be cooked",
> > > > "17.66-31.28": "A man is sitting next to a stove tending the fire, with a child helping nearby",
> > > > "31.28-37.49": "Two little girls are happily eating watermelon in their hands",
> > > > "37.49-64.08": "Several children are sitting on a wall enjoying the scenery outside",
> > > > "64.08-76.50": "After they finish eating, they leave in their respective cars")
> > > >
> > > > **ActivityNet Caption:**
> > > >
> > > > "v_t0ajvfx6dgA": {
> > > >     "duration": 147.26,
> > > >     "timestamps": [
> > > >         [0, 10.31],
> > > >         [17.67, 137.69],
> > > >         [120.02, 125.17]
> > > >     ],
> > > >     "sentences": [
> > > >         "A man in a suit is sitting behind a desk.",
> > > >         "People are playing lacrosse on a field of grass.",
> > > >         "A person in a yellow uniform is standing in front of a net blocking balls."
> > > >     ]
> > > > },
> > > >
> > > > "v_cp52LdlmlUk": {
> > > >     "duration": 129.96,
> > > >     "timestamps": [
> > > >         [0, 129.96],
> > > >         [5.2, 129.96],
> > > >         [17.55, 129.96]
> > > >     ],
> > > >     "sentences": [
> > > >         "People are standing in a room working out.",
> > > >         "They are stepping up and down on small stepping stools.",
> > > >         "They continue working out in the room."
> > > >     ]
> > > > }
> > > >
> > > > To address the concerns and questions, we added the extra experiments and analysis in Appendix H.

---

> ### Comment · Reviewer_Qgjf · 2024-11-25
> **Official Comment by Reviewer Qgjf**
>
> Thank you for the further feedback on my review. Below are my responses:
>
> > "In contrast, traditional video caption datasets typically offer a single, high-level description for the entire video."
>
> This statement is inaccurate. Datasets like HD-VILA-100M and Panda-70M also segment videos into multiple clips and provide caption annotations for each segment. Regarding the Youku Dense Caption annotations, the provided format is same as that of regular video caption datasets. For instance, the annotation format in Panda-70M (as in the csv file: https://drive.google.com/file/d/1BZ9L-157Au1TwmkwlJV8nZQvSRLIiFhq/view) shows the same format as in Youku Dense Caption.
>
> For ActivityNet, it is a dense caption dataset because the annotation includes temporal-related information. Taking the provided sample "v_cp52LdlmlUk" as an example, it has the structure: "People doing something" → "Then doing another thing" → "Then doing yet another thing". This information is what differentiates dense caption datasets from regular video caption datasets.
>
> > "We have also provided additional test results on the Youku-mPLUG validation set, utilizing Vatex and Youku Dense Caption for training"
>
> Please include this table and details about the experimental setup in the revised version of the paper. It would also be helpful to provide more comprehensive results, such as using different combinations of Vatex and Youku Dense Caption as training datasets, similar to what was presented in the original manuscript.

---

> > ### Author Response · Authors · 2024-11-25
> > **Rebuttal by Authors**
> >
> > Thank you for the further feedback.
> >
> > First, the average duration of the videos in our dataset is 85.6 seconds, and each video has an average of 9.9 annotation texts, with an average duration of each annotation being 8.1 seconds. Additionally, we provide numerous examples containing rich temporal-related information in Figure 1 and Appendix G, as we previously indicated in our rebuttal with the "1478912125" example from Youku Dense Caption, which perfectly fits the format "People doing something" → "Then doing another thing" → "Then doing yet another thing."
> >
> > Our dataset is indeed a dense video caption dataset.
> >
> > Second, we have already added all the experiments from the rebuttal in the paper's revised version, particularly in Appendix H addressing your questions (We have made this clear in the previous rebuttal). Using different combinations of Vatex and Youku Dense Caption might be helpful, but it is not necessary, as the ablation experiments and Appendix H have fully demonstrated that our dataset can bring significant improvements to existing benchmarks.

---

> > > ### Author Response · Authors · 2024-11-27
> > > **Look forward to your post-rebuttal feedback!**
> > >
> > > Dear Reviewer Qgjf,
> > >
> > > Thanks again for your insightful suggestions and comments. Since the deadline of discussion is approaching, we are happy to provide any additional clarification that you may need.
> > >
> > > In our previous response, we have carefully studied your comments and made detailed responses summarized below:
> > >
> > > - Restate the similarities and differences with some existing datasets, such as Panda-70M and ActivityNet Caption, indicating that we are Dense Video Caption dataset.
> > >
> > > - Compared the average length of annotations in several existing manually annotated datasets, showing that our dataset's annotations are longer and more detailed than most existing datasets.
> > >
> > > - Supplemented with dense captioning experiments on Youku-mPLUG, addressing the dataset distribution issue, and demonstrated that our dataset is of higher quality, adding more contributions to existing benchmarks.
> > >
> > > We hope that the provided experiments and the additional explanation have convinced you of the contributions of our submission.
> > >
> > > Please do not hesitate to contact us if there's additional clarification or experiments we can offer. Thanks!
> > >
> > > Thank you for your time!
> > >
> > > Best, Authors

---

> > > > ### Author Response · Authors · 2024-11-29
> > > > **Look forward to your further feedback!**
> > > >
> > > > Dear Reviewer Qgjf,
> > > >
> > > > Thanks again for your insightful suggestions and comments. Since the deadline of discussion is approaching, we are happy to provide any additional clarification that you may need.
> > > >
> > > > We've addressed all the points you raised and hope our responses have resolved your concerns. We would highly appreciate it if you could kindly reconsider the rating of our work given the initial score as 3.
> > > >
> > > > Please do not hesitate to contact us if there's additional clarification or experiments we can offer. Thanks!
> > > >
> > > > Thank you for your time!
> > > >
> > > > Best, Authors

---

### Official Review · Reviewer_Va4c · 2024-10-30

**Soundness:** 4
**Presentation:** 3
**Contribution:** 4
**Rating:** 8
**Confidence:** 4

**Summary:**

This paper details a novel Chinese language based dense video caption dataset.

**Strengths:**

+ This work collected a new Chinese based dense video captioning dataset (named Youku Dense Caption). The dataset has several benchmarks for Chinese video-language tasks, including retrieval, grounding, and generation tasks.
+ This allow developer and researcher to train multmodal foundation model with a fair benchmark.
+ This submission has validate the impact of large scale dataset on existing multimodal model. Providing empirical evidence of the advantage of rich data in the context of Chinese language.

**Weaknesses:**

- The proposed dataset's video are evenly sampled from the Youku-mPLUG dataset based on dedicated (sub)categories. So the assumption is that the licensing should not be an issues. To properly handle the copyright concerns, please details the licensing terms for the Youku-mPLUG dataset, and discuss the coverage of usage right, redistribution policies, and any restriction.

**Questions:**

- Has the author consider to release the English caption of the proposed dataset? In my opinion, this will broaden the impact of this dataset and benefit more downstream tasks. Please confirm if this is a a planned future work, as well as to discuss the plan for creating high-quality English translation of the captions. Further analysis on the English translation with open-source tool in Fig 1, it is clear that it requires professional proofreading or checked by someone (crowdsourcing?) who are fluent in both Chinese and English language.

- This paper should provide addition details related to the annotation progress. Please provide detilas on:
1. The total number of human annotators involved.
2. The qualifications or expertise of the annotators (e.g., native Chinese speakers, etc.) and how are them recruited.
3. The cost and time spent on annotation per video.
4. The total number of annotations per annotator and the overall duration.
5. Any quality control measures sued during the annotation process.

---

> ### Author Response · Authors · 2024-11-15
> **Rebuttal by Authors**
>
> Thank you for your valuable feedback. We appreciate the opportunity to address your concerns and provide additional clarification.
>
> Regarding the weakness you mentioned, we sincerely apologize for any oversight. Our open-source license is the same as that of Youku-mPLUG[1], as we mentioned in the ETHICS STATEMENT section. This section comprehensively addresses the open-source license and potential impacts of our work. The dataset will not be used for commercial purposes, will be properly attributed, and will be shared under the same conditions.
>
> Concerning question 1, we acknowledge your point about an English version of the Youku dense caption dataset. Initially, our focus was on the Chinese language community, given the abundance of high-quality, influential datasets already available in English. However, we appreciate your suggestion and will seriously consider releasing an English version in the future. If we proceed with this, we would likely utilize more advanced APIs such as GPT-4 to ensure the highest possible quality of the translated data.
>
> We're glad to provide additional details about our annotation process:
>
> 1. Our annotation team consisted of 1 annotation leader and 10 annotators. The leader was responsible for aligning annotation standards, conducting trial annotations, training annotators, and quality control. The annotators focused on specific annotation tasks and data revisions.
>
> 2. We engaged a professional data annotation agency. The annotation leader had at least 3 years of experience in managing annotation tasks, while the annotators had a minimum of 3 months of experience in CV-related data annotation. All team members were native Chinese speakers.
>
> 3. The annotation cost was 0.63 RMB per valid clip caption, with total video costs varying based on the number of valid slices. The entire process took 2 months, with efficiency improving over time as annotators became more proficient.
>
> 4. While we don't have detailed information about the internal labor distribution within the annotation agency, we understand from our interactions with the annotation leader that the workload was generally evenly distributed among annotators.
>
> 5. The annotation leader conducted quality checks on 10% of each batch of annotations. A 97% pass rate was required for batch approval; otherwise, the entire batch was returned for re-annotation. In the early stages, we also performed secondary reviews of the data to ensure alignment with annotation standards.
>
> We hope this additional information addresses your concerns and provides a clearer picture of our rigorous annotation process. We remain committed to maintaining high standards of data quality and transparency in our research.
>
> [1] Xu, Haiyang, et al. "Youku-mplug: A 10 million large-scale chinese video-language dataset for pre-training and benchmarks." arXiv preprint arXiv:2306.04362 (2023).

---

> > ### Comment · Reviewer_Va4c · 2024-11-27
> >
> > I appreciate the authors' efforts in explaining the annotation detail and other related licensing details.
> >
> > I agree that influential datasets are available in English, but I encourage making the dataset multilingual if possible. I believe this may provide some new research direction (although baby steps in this direction). Given a video input, how will the same content be captioned in different languages, and how different cultural backgrounds view the content differently? This may lead to many interesting directions in the future.
> >
> > Currently, the new dataset is a Chinese language-based annotation. It can serve as a common benchmark for validating different models. While this may provide unique research, one may argue that it does not offer new insights to the research community. Despite that, it is a welcomed effort.

---

### Official Review · Reviewer_QnCD · 2024-10-30

**Soundness:** 3
**Presentation:** 3
**Contribution:** 3
**Rating:** 6
**Confidence:** 4

**Summary:**

The paper presents Youku Dense Caption, the largest publicly available dataset for Chinese dense video captioning, comprising 31,466 videos annotated with 311,921 captions. Collected from the Youku video platform, this dataset addresses the lack of high-quality Chinese video captioning datasets and promotes advancements in Chinese multi-modal research. It provides benchmarks for key tasks such as retrieval, grounding, and generation, and extensive experiments demonstrate its effectiveness on state-of-the-art multi-modal models. The dataset’s scale and quality make it a valuable resource for future research in video-language understanding.

**Strengths:**

Youku Dense Caption contains 31,466 short videos annotated with 311,921 captions, making it the largest dataset for fine-grained Chinese video descriptions.It addresses the scarcity of high-quality Chinese dense video captioning datasets, promoting advancements in Chinese multi-modal models and video-language research.The dataset establishes benchmarks for key video-language tasks such as retrieval, grounding, and generation, with extensive experiments demonstrating the utility of the dataset on state-of-the-art multi-modal models.

**Weaknesses:**

1. Lack of Caption Diversity and Detail: In Figure 1, the captions for different segments of Video ID 1070344446 show high similarity with little distinction, lacking sufficient variability. Additionally, the descriptions are relatively simple and do not provide background information about the visual content.
2. Potential Hallucination in Captions: In the D. Implementation Details of Baselines section, the authors mention that they convert videos to 320p resolution and remove the audio component. However, in Figure 1, the second frame of Video ID 1192027222 shows the caption: “The old lady boasts that young women who work hard at chopping can do it.” It is difficult to determine solely from the visual content that the old lady is boasting, raising concerns about the potential for hallucinated captions, especially for those tied to audio-related information.

**Questions:**

1. Vocabulary Statistics and Comparison: Can the authors provide vocabulary statistics and a comparison with other datasets? The captions appear to contain a high degree of repetition.
2. Threshold for Average Self-BLEU: Why was the threshold for Average Self-BLEU set to 0.15?

---

> ### Author Response · Authors · 2024-11-15
> **Rebuttal by Authors**
>
> Thank you for your thorough review and valuable feedback on our paper. We greatly appreciate your insights and would like to address your concerns as follows:
>
> Regarding Weakness 1:
>
> We understand your concern about the similarity of annotation texts. Figure 1 only shows examples of the original annotations in the dataset, which are in their raw form before being filtered by the algorithm proposed in Section 4. This is not the actual data we used in our experiments. While it's challenging to completely avoid similar texts during the annotation process without significantly increasing difficulty, we've implemented rigorous measures to address this issue. As shown in Algorithm 1, we use the state-of-the-art Chinese embedding model (xiaobu_embedding[1]) to filter out texts with over 90% similarity. This ensures that our final dataset doesn't contain highly similar texts, thus maintaining diversity and quality.
>
> Regarding Weakness 2:
>
> We apologize for any confusion our previous statement may have caused. In the soon-to-be-released dataset, we will retain all original content, including audio and high-resolution video quality. The removal of audio in our experiments was solely due to the limitations of the models used, which don't process speech. This decision doesn't affect the integrity or value of the dataset itself.
>
> Regarding Question 1:
>
> We appreciate your insightful question. As our dataset is the first Chinese dense video dataset, it's indeed challenging to find a directly comparable dataset for vocabulary analysis. However, we've provided a comparative statistical analysis of the original annotation text and the processed data in Section 4, as shown in the table. This comparison demonstrates the changes in word frequency distribution, addressing the issue of repetition.
>
> | Rank | Original Annotation Word | Original Freq. | Processed Word | Processed Freq. |
> |------|------------------------------|----------------------------|--------------------------------|------------------------------|
> | 25   | 一边 (One side) | 0.48611 | 黄色 (Yellow) | 0.45717 |
> | 75   | 男士 (Gentleman) | 0.19027 | 西装 (Suit) | 0.18388 |
> | 125  | 手指 (Finger) | 0.10943 | 狮子 (Lion) | 0.11035 |
> | 175  | 西服 (Suit) | 0.07742 | 一把 (A handful) | 0.08013 |
> | 225  | 外面 (Outside) | 0.06525 | 一台 (One unit) | 0.06621 |
> | 275  | 房间内 (Inside the room) | 0.05288 | 美味 (Delicious) | 0.05445 |
> | 325  | 黝黑 (Dark) | 0.04419 | 牛仔裤 (Jeans) | 0.04496 |
> | 375  | 毛衣 (Sweater) | 0.03774 | 加入 (Join) | 0.03805 |
> | 425  | 安静 (Quiet) | 0.03356 | 赛场 (Arena) | 0.03444 |
> | 475  | 走来走去 (Walk back and forth) | 0.02899 | 长裙 (Long dress) | 0.03001 |
> | 525  | 桌面上 (On the desk) | 0.02641 | 点头 (Nod) | 0.02681 |
> | 575  | 墙边 (By the wall) | 0.02383 | 阳光 (Sunshine) | 0.02424 |
> | 625  | 电动车 (Electric vehicle) | 0.02164 | 愤怒 (Angry) | 0.02248 |
> | 675  | 撕咬 (Tear with teeth) | 0.02023 | 大厅 (Hall) | 0.02063 |
> | 725  | 花白 (Grizzled) | 0.01881 | 花纹 (Pattern) | 0.01928 |
> | 775  | 剪刀 (Scissors) | 0.01752 | 轮胎 (Tire) | 0.01836 |
> | 825  | 工作服 (Work clothes) | 0.01642 | 笔直 (Straight) | 0.01702 |
> | 875  | 开着车 (Driving) | 0.01546 | 慢悠悠 (Slowly) | 0.01588 |
> | 925  | 卡其色 (Khaki) | 0.01443 | 头顶 (Top of head) | 0.01475 |
> | 975  | 刷子 (Brush) | 0.01353 | 展厅 (Exhibition hall) | 0.01403 |
>
>
> | Indicator | Original Annotation | Processed Data |
> |------------------|------------------------|--------------------------|
> | Range of Word Frequencies | 0.4726 | 0.4431 |
> | Standard Deviation of Word Frequencies) | 0.1928 | 0.1824 |
>
>
> Regarding Question 2:
>
> Your question about the self-BLEU threshold is crucial. As mentioned in Section 4.2, our choice of self-BLEU threshold aims to balance data diversity and suitability for generation and moment retrieval tasks. After manually observing hundreds of video samples under different threshold settings, we determined that a self-BLEU threshold of 0.15 provides the optimal balance between text distinctiveness and task applicability.
>
> We hope these responses address your concerns adequately. If you have any further questions or require additional clarification, please don't hesitate to ask. We're committed to improving our research based on your valuable feedback.
>
> [1] https://huggingface.co/lier007/xiaobu-embedding-v2

---

> > ### Comment · Reviewer_QnCD · 2024-11-16
> >
> > Great rebuttal! As a reviewer, I appreciate your efforts.
> >
> > So for W1, would you mind to provide some visualization of actual data you used in our experiments? I think the Figure 1 may be a little confused with similar texts.
> >
> > For Q1, please update the context in the appendix and indicate the page numbers in your follow-up response.
> >
> > For Q2, would you mind offering some analysis about the self-BLEU threshold? For example, how will the filtered dataset under different threshold influence downstream performance? I know that it may not be possible to complete this part of the experiments in a short time for the rebuttal. You don’t need to provide me with exact statistics; I’m just looking for a rough estimate.

---

> > > ### Author Response · Authors · 2024-11-22
> > > **Rebuttal by Authors**
> > >
> > > Thank you for your comments and suggestions. We appreciate the opportunity to address your concerns and provide additional clarification.
> > >
> > > Regarding W1, we have added more visual examples in Appendix G of the updated paper to better illustrate our dataset. We hope this will help alleviate your concerns and provide a clearer understanding of our dataset.
> > >
> > > In response to Q1, we added a new "Vocabulary Statistics" section in Appendix E (line 938), which analyzes and presents the changes in vocabulary frequency distribution before and after quality filtering. Our results demonstrate that post-filtering, the vocabulary distribution becomes more balanced, significantly reducing the issue of word repetition.
> > >
> > > Addressing Q2, we added a discussion and analysis of the self-BLEU threshold selection in Appendix F. This includes examples of annotations with self-BLEU scores around the 0.15 threshold, highlighting the distinctions between them. As shown in Tables 10 and 11 of the paper, when the self-BLEU score is slightly above 0.15, certain words tend to appear repeatedly across all annotation texts. In contrast, when the score is slightly below 0.15, repeated words are found only in some of the annotation texts. Based on these observations, we infer that a self-BLEU threshold of 0.15 effectively mitigates the overly frequent occurrence of certain phrases in the annotation texts.
> > >
> > > We believe these additions and clarifications address your concerns comprehensively. We are grateful for your feedback, as it has helped us improve the quality and clarity of our paper. Please let us know if you have any further questions or if there's anything else we can elaborate on.

---

> > > > ### Comment · Reviewer_QnCD · 2024-11-22
> > > >
> > > > Thanks for your information. Good luck for the rest of your rebuttal.

---

### Official Review · Reviewer_C5mS · 2024-11-03

**Soundness:** 3
**Presentation:** 3
**Contribution:** 2
**Rating:** 6
**Confidence:** 3

**Summary:**

This paper introduces Youku Dense Caption, a large-scale Chinese dense video captioning dataset. Dataset addresses the scarcity of high-quality Chinese video captioning resources, containing 31,466 short videos with 311,921 Chinese captions. A strategies is proposed to improve benchmark quality by filtering out redundant or low-quality annotations. The authors establish several benchmarks for Chinese video-language tasks and conduct extensive experiments demonstrating the dataset's utility and potential for research. They also discuss challenges related to the linguistic and cultural differences between Chinese and English video data.

**Strengths:**

1. A Chinese video captioning dataset is proposed to fill the research gap in the Chinese community for video captioning data.
2. A embedding-based similarity and a Non-Maximum Suppression method is used to set up a Chinese PRVR benchmark that effectively reduces annotation redundancy.
3. The work reduces redundancy in video captioning and grounding by filtering out videos with high self-BLEU scores and minimal scene changes,  which is measured through color histogram correlation, ensuring a diverse and representative dataset.

**Weaknesses:**

1. The statement in the section “Chinese Characteristics” seems unclear. The English translation of the so-called “fine-grained Chinese captions” could also serve as fine-grained English captions in an English-language context. For me, only the localized data part is valuable, as it highlights a major difference between Chinese and English video captions. Adding more data and statistics to support this distinction would strengthen the paper.
2. In the experiment, when translated back to Chinese. It's kind of blur that which attributes of Chinese dataset lead to the poor performance, the analysis failed to state clearly about the language differences between Chinese and English.
3. In ablation study, mixing of different datasets only strike a balance between different tasks but failed to achieve idealized performance across different tasks. And the best performance comes from larger data scale rather than data distribution and video-caption pair.
4. Overall, the dataset serve as a valuable data source for Chinese community in video caption domain, but the value and key attributes of the dataset remain unclear and is not fully proved by the experiment.

**Questions:**

see weakness.

---

> ### Author Response · Authors · 2024-11-15
> **Rebuttal by Authors**
>
> We sincerely appreciate your thorough review and insightful comments. We would like to address your concerns as follows:
>
> Regarding Weakness 1:
>
> We acknowledge that our use of "fine-grained Chinese captions" may be misleading. A more accurate description would be "Chinese dense captions", as we divide videos into fine-grained segments and provide annotations for each segment. We will revise this terminology in the relevant sections to ensure clarity.
>
> Regarding Weakness 2:
>
> We have attempted to elaborate on the reasons for the discrepancies between Chinese and English annotations in Section 3.2.2. This includes differences in word usage (e.g., "切" vs. "剁"), cultural nuances (such as tea ceremonies), and unique vocabulary (like "新闻联播"). These factors likely contribute to the suboptimal performance of translated data in our experiments. We believe these distinctions underscore the importance of a native Chinese dataset.
>
> Regarding Weakness 3:
>
> In Section 6, we demonstrate the value of our dataset through experiments on video captioning and retrieval tasks. By augmenting the Vatex dataset with our data, we achieved state-of-the-art performance on both the Vatex[1] validation set and our Youku dense caption benchmark. Similarly, for video retrieval, gradually incorporating our dataset into training with the Youku-mPLUG dataset led to state-of-the-art results on the Youku-mPLUG[2] validation set. These experiments provide empirical evidence of our dataset's positive impact on existing benchmarks.
>
> Regarding Weakness 4:
>
> We have provided a detailed analysis of the differences between Chinese and English linguistic characteristics in Section 3.2.2, theoretically justifying the necessity of the first Chinese dense video dataset. In Section 5, experiments across retrieval, generation, and grounding tasks show that Chinese-specific models like Chinese-CLIP[3] outperform models trained on translated English data, further validating our dataset's importance. Our ablation studies on Vatex and Youku-mPLUG datasets demonstrate that incorporating our dataset improves performance on other benchmarks, substantiating its value.
>
> We hope these clarifications address your concerns and highlight the contributions of our work. We are committed to improving the presentation of our research based on your valuable feedback. If you have any further questions or require additional information, please don't hesitate to ask.
>
>
> [1] Wang, Xin, et al. "Vatex: A large-scale, high-quality multilingual dataset for video-and-language research." Proceedings of the IEEE/CVF international conference on computer vision. 2019.
>
> [2] Xu, Haiyang, et al. "Youku-mplug: A 10 million large-scale chinese video-language dataset for pre-training and benchmarks." arXiv preprint arXiv:2306.04362 (2023).
>
> [3] Yang, An, et al. "Chinese clip: Contrastive vision-language pretraining in chinese." arXiv preprint arXiv:2211.01335 (2022).

---

> > ### Author Response · Authors · 2024-11-22
> > **Rebuttal by Authors**
> >
> > Thank you for your valuable time and effort in reviewing our manuscript. We understand that you must be very busy, and we greatly appreciate the time and effort you have already dedicated to reviewing our work. However, as the rebuttal period is nearing its end, we were wondering if you have had the chance to review our response.
> >
> > If there's any additional information or clarification needed from our side, please don't hesitate to let us know. We are more than happy to provide any further details that might assist in your evaluation.

---

> > > ### Author Response · Authors · 2024-11-25
> > > **Look forward to your post-rebuttal feedback!**
> > >
> > > Dear Reviewer C5mS,
> > >
> > > Thanks again for your insightful suggestions and comments. Since the deadline of discussion is approaching, we are happy to provide any additional clarification that you may need.
> > >
> > > In our previous response, we have carefully studied your comments and made detailed responses summarized below:
> > >
> > > - Terminology Change: "Fine-grained Chinese captions" will be revised to "Chinese dense captions" for accuracy, as segments are annotated.
> > >
> > > - Annotation Discrepancies: Section 3.2.2 discusses differences in word usage, cultural nuances, and unique vocabulary between Chinese and English annotations, highlighting the importance of a native Chinese dataset.
> > >
> > > - Experimental Validation: In Section 6, we show that augmenting the Vatex dataset with our data leads to state-of-the-art performance in video captioning and retrieval tasks.
> > >
> > > - Linguistic Analysis: Section 3.2.2 provides an analysis of differences in linguistic characteristics between Chinese and English, justifying the need for a Chinese dense video dataset.
> > >
> > > - Dataset Impact: Experiments in Section 5 confirm that Chinese-specific models outperform those trained on translated data, and ablation studies demonstrate that our dataset enhances performance on existing benchmarks.
> > >
> > > We hope that the provided experiments and the additional explanation have convinced you of the contributions of our submission.
> > >
> > > Please do not hesitate to contact us if there's additional clarification or experiments we can offer. Thanks!
> > >
> > > Thank you for your time!
> > >
> > > Best, Authors

---

> ### Comment · Reviewer_C5mS · 2024-11-25
> **Reply to author response**
>
> Thank you for your reply and efforts on the rebuttal! My current response is based on the latest version of the paper.
>
> For Weakness 1: Your response and revisions have not addressed my concerns. My main issue is that for English dense caption video datasets, translating these caption into accurate Chinese could also constitute a Chinese video caption dataset. For reference, the sentence in lines 205–207: “For example, the English phrase ‘The woman is cutting vegetables in the kitchen’ translates to ‘女人在厨房里切菜’, where ‘切菜’ can mean ‘cutting vegetables’ or ‘chopping vegetables’, each with different nuances” is difficult to understand as an explanation of so-called Chinese characteristics. Furthermore, examples like “一家人在吃饭，其中一位女子帮他端了一碗酒” (“The family is having dinner, and one of the women brings him a bowl of wine”) can also be considered fine-grained video captions in English; it’s just a different level of fine-grainedness. Claiming this difference stems from different linguistic communities lacks sufficient evidence.
>
> For Weakness 2: My question has been resolved. Thanks for the reply!
>
> For Weakness 3: The introduction of YDC shows limited improvement on the VATEX dataset, and there is a performance drop on the METEOR metric (comparing 100% VATEX + 0% YDC to 100% VATEX + 50% YDC). Additionally, for YDC data, as mentioned in the paper, the OOD setting leads to a performance drop of 13% with the same amount of data. This makes the value of fine-grained captions appear limited, as most of the improvement seems to come from the increased data volume. Moreover, on the 100% Youku-mPLUG dataset, adding 13% YDC data leads to a drop in video-to-text R10, text-to-video R5, and text-to-video R10 metrics. This does not support the claim that “based on the Youku-mPLUG training set, the video retrieval performance steadily improves with the addition of YDC training data.”
>
> For Weakness 4: My concerns regarding this have already been stated under Weakness 1.
>
> From my perspective, I appreciate the value this dataset brings to the Chinese community, but I believe the value primarily lies in the localized data. It would be better to focus more on reducing the gap caused by differences in localized data distribution.

---

> > ### Author Response · Authors · 2024-11-26
> > **Rebuttal by Authors**
> >
> > Thank you for your valuable feedback.
> >
> > As we mentioned in our response to reviewer Qgjf, while advanced language models like ChatGPT could indeed improve Chinese translation quality, the primary motivation behind our dataset extends beyond language differences to address cultural and content disparities between Chinese and foreign contexts. As noted in Section 3.2.2, unique Chinese cultural elements such as tea ceremonies, traditional festivals (enjoying the moon during the Mid-Autumn Festival and paying New Year's greetings on New Year's Day), etc., are rarely found in public English datasets but are common in Chinese contexts. This difference is reflected not only in the distribution of annotation text but also in video content distribution. As highlighted by the widely-used Chinese CLIP, bridging the gap caused by cultural differences in annotation and video content is the core value of our dataset.
> >
> > Regarding weakness 3, different datasets have different distributions, and merely adding equal amounts of data can often lead to performance decline. However, we can compensated for this distribution gap with a larger dataset, thereby achieving better performance. The claim "steadily improves" is indeed inaccurate, and we have corrected this expression in the revised paper.
> >
> > Additionally, as mentioned in our response to reviewer Qgjf, in Appendix H, to avoid the dataset distribution issue, we supplemented experiments by training Qwen2-VL on either the Vatex dataset and the YDC dataset, and then evaluated on the Youku-mPLUG validation set, which is outside the training set distribution. The experimental results indicate that the model trained on our dataset achieves better performance on the Youku-mPLUG validation set, further demonstrating the value of our dataset.
> >
> > | Training Data | BLEU-4 | METEOR | ROUGE-L | CIDEr |
> > |---------------|--------|--------|---------|-------|
> > | vatex         | 3.08     | 12.77     | 19.53      | 29.53    |
> > | Youku Dense Caption          | 3.46     | 13.54     | 22.76      | 31.55    |
> >
> > We hope these responses address your concerns adequately. If you have any further questions or require additional clarification, please don't hesitate to ask. We're committed to improving our research based on your valuable feedback.

---

> > > ### Comment · Reviewer_C5mS · 2024-11-26
> > > **Reply to author response**
> > >
> > > Thank you for your reply! I appreciate your time and effort.
> > >
> > > Regarding weakness 1, I value your explanation of the significance of Chinese culture-specific captions and video content. However, to me, this only corresponds to the last paragraph of Section 3.2.2. The other aspects described as “CHINESE CHARACTERISTICS” are somewhat confusing. Since Section 3.2.2 is meant to highlight the key value of the dataset, the statements in the paper are insufficient to convincingly demonstrate the dataset’s value.
> > >
> > > Regarding weakness 3, my concern lies in whether the observed improvement is due to the dataset’s volume or its high-quality captions. Since nuanced and dense captions are one of the dataset’s major contributions, evidence that the improvement is primarily driven by the dataset scale would strengthen the claim about the value of high-quality captions.

---

> ### Author Response · Authors · 2024-11-27
> **Rebuttal by Authors**
>
> Thank you for pointing out the issues in our paper. We have taken your feedback seriously and have rewritten Section 3.2.2. Specifically, we have condensed the discussion on language translation to one paragraph. Additionally, we have added three paragraphs at the beginning of the section to introduce and describe unique aspects of Chinese culture and explain how these contribute to the differences in data distribution between Chinese and English. At the end of this section, we have summarized our findings, emphasizing that the distinctiveness of our dataset’s Chinese characteristics stems from various cultural contexts, which involve not only differences in annotations but also in video content.
>
> Regarding weakness 3, as discussed in our ablation study, the Vatex dataset and YDC dataset have training sets of similar size for the dense video captioning task. This implies that training on YDC, with a dataset of the same scale, can achieve better results on the Youku-mPLUG validation set, thereby demonstrating the higher quality of our data.
>
> We sincerely appreciate your thorough review and constructive suggestions. We hope these revisions meet your expectations and look forward to your further feedback.

---

> > ### Comment · Reviewer_C5mS · 2024-11-27
> > **Reply to author response**
> >
> > I appreciate the modifications and responses made by the authors, thank you!
> >
> >
> > 1. Regarding Section 3.2.2, the expanded explanation of the uniqueness of the Chinese context makes the value of the dataset much clearer. As for whether two examples are necessary to illustrate this, I hold a reserved opinion. It might introduce some redundancy, but it’s acceptable if length allows.
> >
> > 2. For the second-to-last paragraph in section 3.2.2, I still find it confusing. In the statement, _“For example, in Chinese annotations, terms like ‘剁菜’ (chopping vegetables), ‘切菜’ (slicing vegetables), and ‘砍菜’ (dicing vegetables) is often translated as ‘cutting vegetables’,”_  it seems that the three different Chinese terms correspond to different English captions. However, the following statement says they are often translated as _“cutting vegetables”_ which feels contradictory. This does not seem to illustrate linguistic differences effectively. The two subsequent examples, better demonstrate cultural differences rather than linguistic differences.
> >
> > 3. For culture-specific caption content, I think calculating the proportion of such content in the overall captions and their corresponding English translations in English-caption datasets would provide stronger evidence. I suggest adding such data for support.
> >
> > 4. The performance on the Youku-mPLUG validation set is convincing. However, another concern is that **YDC is evenly sampled from Youku-mPLUG, while the VATEX dataset does not overlap with Youku-mPLUG**. Could this lead to performance drops, as Table 6 indicates, where OOD causes a decline in performance? The additional experiments do not clarify whether deduplication has been performed. Furthermore, based on the results in Table 6, YDC inherently has an advantage over the VATEX dataset as in-domain data. I hope this issue can be effectively addressed, preferably with supporting experimental results.
> >
> > If the above concerns are resolved, I will raise my score. Overall, I appreciate the contribution of this work to the Chinese video annotation community, but I still have concerns about several key aspects of the paper that require clarification.

---

> > > ### Author Response · Authors · 2024-11-27
> > > **Rebuttal by Authors**
> > >
> > > Thank you for your insightful feedback and the opportunity to improve our manuscript. We have addressed each of your points as follows:
> > >
> > > 1. Regarding your first point, we have supplemented Figure 3 with additional examples to better illustrate the characteristics of Chinese culture.
> > >
> > > 2. For your second point, we have reorganized Section 3.2.2 in light of the revised Figure 3, removing the confused elements and adding more detailed descriptions of cultural characteristics.
> > >
> > > 3. On the third point, to best of our knowledge, that datasets like Vatex or ActivityNet Caption do not contain examples with specific descriptions such as Mid-Autumn Festival, mooncakes, New Year, or similar cultural references. Defining and identifying such culturally specific descriptions in a short time  is challenging. However, we have conducted a rough estimation: in the YDC dataset, there are approximately 53 annotations related to the Lunar New Year, 23 related to Mid-Autumn Festival, and 164 related to uniquely Chinese arts like cross-talk. While showcasing Chinese cultural features is a core aspect of our dataset's value, we believe that the high-quality dense captions in non-culture-specific areas are also a significant contribution to the community.
> > >
> > > 4. Addressing your fourth point, as mentioned in line 500 of our paper, we ensured no overlap in video data by deduplicating the Youku-mPLUG data using file hashing and several video features during our experiments. Furthermore, as shown in Table 6, training on an equivalent amount of YDC (13% of YDC) did not yield better results, indicating that YDC is not in-domain data for Youku-mPLUG. Hence, both Vatex and YDC data are out-of-domain for Youku-mPLUG. We assure you that there is no need for concern regarding this issue.
> > >
> > > Lastly, we also acknowledge the scarcity of Chinese video datasets, which limits our ability to conduct related experiments further.
> > >
> > > We appreciate your valuable comments and hope that our revisions address your concerns satisfactorily.

---

> > > > ### Comment · Reviewer_C5mS · 2024-11-27
> > > > **Reply to author response**
> > > >
> > > > Thanks for your revision and effort! Most of my concern is solved right now. The last suggestion is that detailed descriptions of cultural characteristics can be simplified and take better advantage of the figure to showcase the cultural differences. Or maybe include detailed examples in the supplementary material. Dedicating three full paragraphs to introducing Chinese culture in a representation learning conference paper feels excessive.
> > > >
> > > > I will raise my score and wish you the best of luck for the rest of your rebuttal.

---

### Author Response · Authors · 2024-12-04
**General Response to All**

We appreciate all the feedback and suggestions from the reviewers. We are pleased that the reviewers recognize our work and acknowledge our efforts to address their concerns.

--------------

Here, we would like to highlight and clarify the **Contributions of Youku Dense Caption**:

1. Why is proposing a new Chinese video captioning dataset a core contribution?
   - The Youku Dense Caption dataset addresses the lack of high-quality Chinese video captioning datasets, being the largest of its kind and promoting advancements in multi-modal models and video-language research.

2. How do the technical methods enhance dataset effectiveness and reduce redundancy?
   - By using embedding-based similarity, Non-Maximum Suppression, and algorithms like filtering videos with high self-BLEU scores and minimal scene changes, this work reduces redundancy in video captioning and grounding, ensuring a diverse and representative dataset.

3. How does the dataset advance research in video-language tasks?
   - The dataset provides benchmarks for tasks like retrieval, grounding, and generation, and experiments on different tasks and existing benchmarks show the advantages of our dataset for training multimodal models in Chinese.

--------------

We duly address the reviewers' concerns in the individual responses.

Here, we summarize the **Key Issues and Clarifications** for addressing the concerns in the revision.

  - Chinese Characteristics: Reviewers questioned the understanding of "Chinese characteristics" in our dataset. We addressed this by providing additional examples and explanations to highlight its significance. (Section 3.2.2)
  - Experimental Methods and Data Gain: Concerns were raised about the effectiveness of the experiments and the dataset's contribution to different tasks. We validated these contributions through ablation studies. (Section 6 Ablation Study)
  - Data Distribution and Validation: To address concerns about data distribution, we provided additional experiments to ensure the reliability of the results. (Appendix H)
  - Diversity and Hallucination Issues: Reviewers were concerned about diversity and hallucination in the dataset. We alleviated these concerns by providing more examples. (Appendix G)
  - Vocabulary Statistics: Reviewers requested detailed vocabulary diversity and generation quality analysis, which we provided. (Appendix E and Appendix F)

--------------

Finally, we sincerely appreciate reviewers for their valuable suggestions, which have significantly helped us enhance the quality of our work. Additionally, we regret that reviewer Qgjf did not have enough time to thoroughly review our manuscript and participate in the rebuttal discussion.

---

### Meta-Review · Area_Chair_wGY7 · 2024-12-16

**Metareview:**

This paper presents Youku Dense Caption, a large-scale Chinese dense video captioning dataset. In the rebuttal, the authors effectively addressed several concerns, including clarifying the key value of the dataset of presenting “Chinese characteristics”, and resolving questions related to cross-dataset evaluation, copyright issues, dense captions and caption quality. As a result, all four reviewers unanimously recommended acceptance. The AC agrees and the authors should further revise the final version according to reviewers’ comments.

**Additional Comments On Reviewer Discussion:**

The paper has initially received mixed ratings.  In their rebuttal, the authors have well addressed several concerns, including clarifying the key value of the dataset of presenting “Chinese characteristics”, and resolving questions on cross-dataset evaluation, copyright issues, dense captions and caption quality. As a result, all four reviewers unanimously recommended acceptance.

---

### Decision · Program_Chairs · 2025-01-22

Accept (Poster)